# Wnt activation disturbs cell competition and causes diffuse invasion of transformed cells through NF-κB-MMP21 pathway

Kazuki Nakai[1], Hancheng Lin[1], Shotaro Yamano [2], Shinya Tanaka[3], Sho Kitamoto [4], Hitoshi Saitoh[5], Kenta Sakuma[1], Junpei Kurauchi[1], Eilma Akter[1], Masamitsu Konno[1], Kojiro Ishibashi[6], Ryo Kamata [5], Akihiro Ohashi [5], Jun Koseki [7], Hirotaka Takahashi[8], Hideshi Yokoyama[9], Yukihiro Shiraki [10], Atsushi Enomoto [10], Sohei Abe[11], Yoku Hayakawa [11], Tetsuo Ushiku[12], Michihiro Mutoh[13], Yasuyuki Fujita[3] & Shunsuke Kon [1] ✉

Normal epithelial cells exert their competitive advantage over RasV12-transformed cells and eliminate them into the apical lumen via cell competition. However, the internal or external factors that compromise cell competition and provoke carcinogenesis remain elusive. In this study, we examine the effect of sequential accumulation of gene mutations, mimicking multi-sequential carcinogenesis on RasV12-induced cell competition in intestinal epithelial tissues. Consequently, we find that the directionality of RasV12-cell extrusion in Wnt-activated epithelia is reversed, and transformed cells are delaminated into the basal lamina via non-cell autonomous MMP21 upregulation. Subsequently, diffusively infiltrating, transformed cells develop into highly invasive carcinomas. The elevated production of MMP21 is elicited partly through NF-κB signaling, blockage of which restores apical elimination of RasV12 cells. We further demonstrate that the NF-κB-MMP21 axis is significantly bolstered in early colorectal carcinoma in humans. Collectively, this study shows that cells with high mutational burdens exploit cell competition for their benefit by behaving as unfit cells, endowing them with an invasion advantage.

Recent advances in cancer genomics have revealed that transformed cells carrying oncogenic insults are frequently produced in tissues[1,2]. However, living organisms implement self-defense mechanisms to suppress oncogenesis, with cell competition being one of the tumor-surveillance systems to remove newly emerging transformed cells[3–6]. In addition to its anti-tumorigenic role, it has become increasingly apparent that divergent physiological processes such as tissue repair, stem cell maintenance, and even aging, are governed, at least partially, by cell competition across a range of epithelial tissues[7–9]. In 1975, Morata et al. discovered that ribosomal-deficient mutants underwent apoptosis when they coexisted with normal epithelial cells in the

imaginal wing disc of *Drosophila*[10]. Since this pioneering study, it has been observed in *Drosophila* that suboptimal cells with aberrant cell polarity, metabolism, membrane trafficking, and other characteristics are eliminated through competitive interactions[7,11–13]. Furthermore, recent studies have uncovered that cell competition is highly conserved in mammals, as exemplified by studies using mammalian experimental models[14–16]. In a previous study, we established a cell competition mouse model by mating LSL-RasV12-*IRES*-eGFP mice with Cre-ERT2 transgenic lines. In this model, RasV12-transformed cells are produced in a mosaic pattern within epithelial layers by the administration of low-dose tamoxifen. Using this model, we demonstrated that

RasV12-transformed cells residing between normal cells are apically extruded via cell competition and are excreted along with other body waste[17]. This model serves as a valuable platform for studying the apical elimination of transformed cells in vivo and is well-suited for functional analysis of cell competition in a physiological setting[18–21]. To date, a series of studies have reported the apical extrusion of RasV12 in several mouse organs, including small intestine[17], stomach[20], pancreas[19], and lung[18]. These findings indicate that the removal of transformed cells through cell competition is an innate defense system orchestrated among epithelial cells, which suppresses the accumulation of mutated lineages and reduces the risk of oncogenesis.

In general, progressive accumulation of genetic aberrations is associated with the onset of carcinogenesis[22,23]. For instance, in human colorectal cancer, the transition from normal epithelial cells to malignant cells follows a stepwise series of genetic mutations, typically involving *APC*, *Ras*, and *p53*, in a specific order[24,25]. Yet, the precise mechanism that determines the sequence of these genetic alterations remains unidentified. Moreover, the impact of accumulated oncogenic insults on cell competition is unclear. This caused us to wonder whether genetic mutations might influence the behavior of transformed cells in a competitive environment. We reasoned that it might be possible to exploit human familial adenomatous polyposis (FAP) as a model to evaluate functional perturbation of cell competition during multi-sequential carcinogenesis. Given that mutations in the *APC* tumor suppressor gene are the most prevalent initiating insults in colorectal cancer[26], we engineered a cell competition mouse model to sustain infrequent somatic activation of *Ras* in *APC*-mutated epithelia in a manner reminiscent of FAP and investigated the fate of transformed cells. In this study, we demonstrate that *APC*-ablation leads to a malfunction of cell competition, converting this normal homeostatic mechanism into a potential driver of tumor progression.

## Results

### Aberrant Wnt activation disturbs apical extrusion and potentiates non-cell autonomous diffuse invasion of RasV12-transformed cells into the basal lamina

To examine the effect of *APC* deficiency on the apical extrusion of RasV12-transformed cells, we utilized *APC^Min^* mice, which harbor a heterozygous loss-of-functional mutation in the *APC* gene[27]. Quantitative real-time PCR (q-PCR) analysis using intestinal epithelial cells of *wild-type*- or *APC^Min^*-derived crypt organoid cultures confirmed a significant elevation in the expression of Wnt-targeted genes in *APC^Min^* mice compared to *wild-type* mice (Supplementary Fig. 1a). In order to somatically activate the H-Ras protein in a small subset of intestinal cells in the context of APC ablation, *APC^Min^* mice were bred with Villin-Cre^ERT2^ mice and *DNMT1*-CAG-*loxP*-STOP-*loxP*-HRas^V12^-*IRES*-eGFP mice (referred to as *APC^Min^-Villin-RasV12* mice) or CAG-*loxP*-STOP-*loxP*-eGFP mice (*APC^Min^-Villin-GFP* mice). While Villin-Cre^ERT2^ mice recombine Cre-responsive elements in both differentiated cells and Lgr5+ stem cells[28], low-dose, tamoxifen-dependent, mosaic expression of transgenes was primarily restricted to terminally differentiated enterocytes, but not occur in Olfm4-positive stem cells, Wheat Germ Agglutinin (WGA)-positive paneth and goblet cells, or DCLK1-positive tuft cells (Supplementary Fig. 1b). This selective expression pattern is likely due to the higher tamoxifen sensitivity of differentiated cells compared to stem cells.

We next examined the fate of newly emerged RasV12-transformed cells in the intestinal tracts of 6–10-weeks-old mice, 3 days after tamoxifen administration. This age group was chosen to avoid aberrant activation of Wnt signaling caused by loss of heterozygosity (LOH) of the *APC* locus in older mice[29]. As a result, a substantial number of RasV12-expressing cells surrounded by normal epithelial cells were apically eliminated in *Villin-RasV12* mice. Interestingly, we observed a small fraction of transformed cells (1.87 ± 0.23%) undergoing basal delamination, with nuclei oriented toward the basal lamina (Fig. 1a, b).

These cells were defined as basally extruded cells. Strikingly, the percentage of cells for which the above criteria were met was significantly higher in *APC^Min^-Villin-RasV12* mice (7.67 ± 1.03%) compared with *Villin-RasV12* mice, while GFP-expressing cells of *APC^Min^-Villin-GFP* mice were either apically or basally extruded no more often than their neighbors (Fig. 1a, b). Moreover, the basally extruded APC^Min^/RasV12-transformed cells penetrated the basement membrane and invaded the surrounding stromal tissues, where they expanded (Supplementary Fig. 2a). Importantly, these APC^Min^/RasV12 cells were negative for Ki-67 staining, a surrogate marker for cell proliferation, ruling out the possibility of growth-stimulated overcrowding as the cause of their delamination (Supplementary Fig. 2b). These findings collectively suggest that Wnt activation disrupts cell competition, resulting in promoted basal extrusion of RasV12-transformed cells in vivo. Furthermore, basal extrusion of APC^Min^/RasV12 cells occurred with a similar frequency throughout the small intestine, from duodenum to ileum (Supplementary Fig. 3a, b). Consistent with the in vivo observations, experiments using intestinal organoids also revealed basal extrusion of APC^Min^/RasV12 cells under the condition of low-dose tamoxifen, while that of APC^Min^/GFP or RasV12-single mutated cells was rarely observed (Fig. 1c, d). Three-dimensional modeling of APC^Min^/RasV12-transformed organoids highlighted elongating, protruding transformed cells that were basally exterminated (Fig. 1e). Importantly, when RasV12 mutants were predominantly induced in *APC^Min^* organoids using high-dose tamoxifen, the frequency of both apical and basal extrusion was drastically reduced in various cluster-forming transformed cells, in a density-dependent fashion (Fig. 1f, g). These results emphasize that the presence of surrounding *APC*-ablated cells is necessary for APC^Min^/RasV12 cells to be expelled from the epithelia, indicating that extrusion of transformed cells occurs non-cell autonomously.

To validate that the effect of APC mutation on cell competition is primarily driven by Wnt activation, we employed a tissue injury model[30]. Treatment with indomethacin resulted in significant damage to the small intestinal tissues, resulting in a gradual reduction in body weight. However, the mice began to recover approximately 3 days after the injury (Supplementary Fig. 4a). At 8 days post-injury, we observed a marked elevation in Wnt activity (Supplementary Fig. 4b). Importantly, basal extrusion of RasV12-transformed cells was significantly enhanced compared to non-injured mice, providing further evidence that Wnt activation is the underlying cause of basal delamination of transformed cells in a competitive environment (Supplementary Fig. 4c, d).

### *APC^Min^-Villin-RasV12* mice develop invasive carcinomas

We then tracked the fate of APC^Min^/RasV12-transformed cells as they invaded the lamina propria long after RasV12 expression. At 21 days after tamoxifen administration, both *Villin-RasV12* mice and *APC^Min^-Villin-GFP* mice displayed no overt phenotypes, with GFP-expressing cells having disappeared due to the rapid turnover of intestinal epithelial cells (Fig. 2a). In marked contrast, *APC^Min^-Villin-RasV12* mice succumbed to intestinal transformation, with GFP-positive APC^Min^/RasV12 cells being confined in the stroma of upper villi (Fig. 2a). Notably, no microscopically detectable architectural deformities in the form of benign adenomatous tumors (adenoma) were observed in proximity to tumors. This implies that cancer cells were generated directly from the normal mucous membrane through the diffuse infiltration of transformants. Furthermore, the stromal cell nests appeared histopathologically undifferentiated, lacking forming glandular ducts, being different from signet-ring cell carcinomas or neuroendocrine neoplasms, and most nearly resembling diffuse-type carcinomas observed in human[31,32]. By 36 days, APC^Min^/RasV12 cells had further invaded the submucosa and even the muscle layer (Fig. 2b). Notably, these cells invaded cohesively, as evidenced by the sustained expression of E-cadherin in the invasive cells (Fig. 2b).

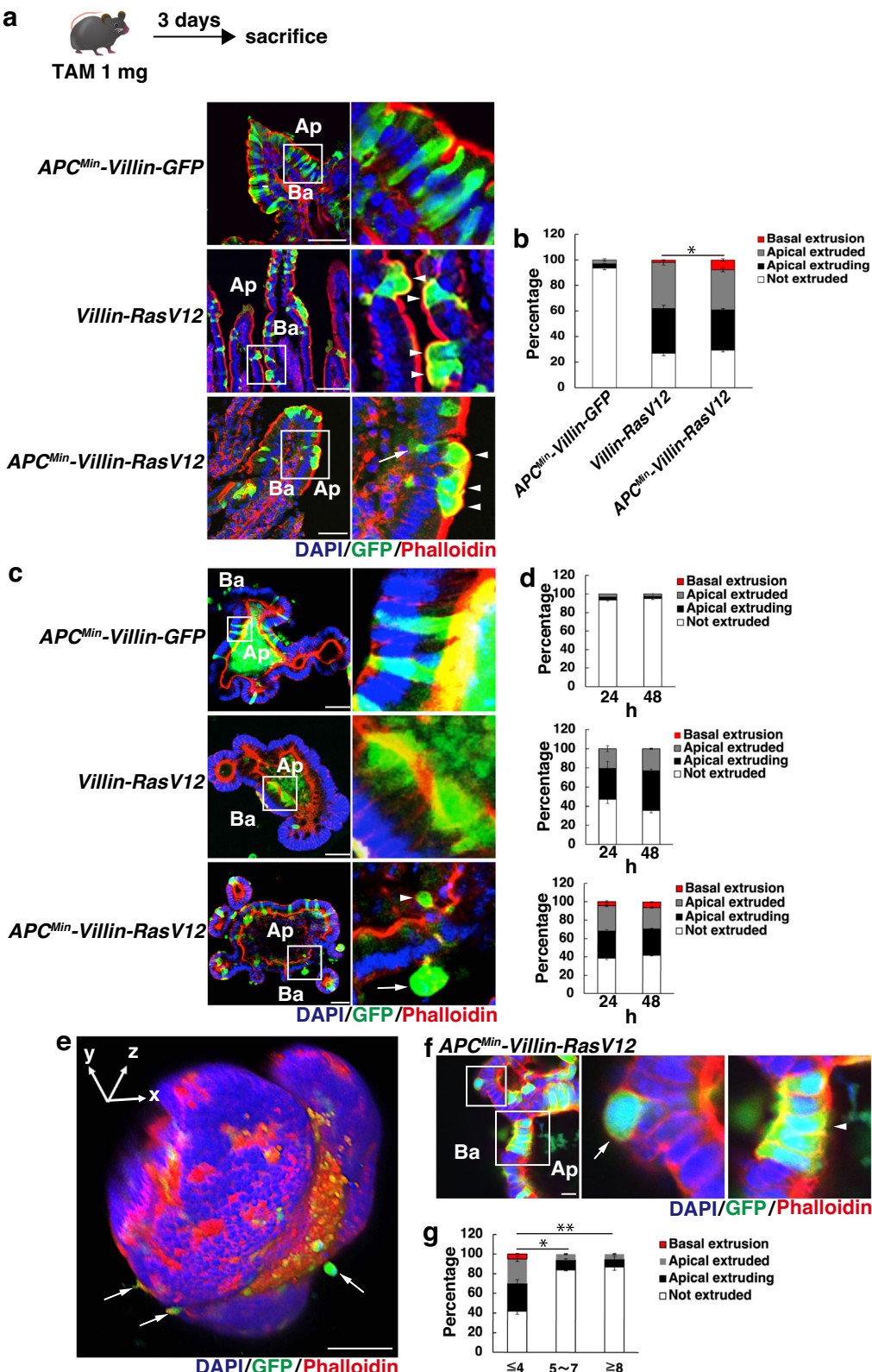

These findings prompted us to thoroughly characterize the properties of APC[Min]/RasV12-malignant cells by evaluating the expression of several markers (Supplementary Fig. 5a). APC[Min]/RasV12 carcinoma cells did not show positive staining for AB-PAS. Additionally, they expressed Sox9 (an intestinal progenitor marker) at the lowest level while maintaining the CDX2 expression (an intestinal lineage marker). In contrast, typical adenoma cells produced in elderly APC[Min]-

Villin-GFP mice displayed the opposite expression pattern, being positive for AB-PAS staining, exhibiting high levels of Sox9 expression, and reduced CDX2 expression. Immunostaining for Ki-67 demonstrated that carcinoma cells showed moderate growth stimulation, in sharp contrast to the highly proliferative status of adenoma cells. This would reflect a stress-resistant status of APC[Min]/RasV12 cells to survive in a harsh stromal environment. Furthermore, APC[Min]/RasV12 cells did

**Fig. 1 | APC^Min/RasV12-transformed cells are basally delaminated in a non-cell autonomous fashion. a, c, e, f** Confocal microscopic images of intestinal villi (**a**) or intestinal organoids (**c**, **e**, **f**) from *APC^Min-Villin-GFP*, *Villin-RasV12*, or *APC^Min-Villin-RasV12* mice. In **a**, *APC^Min-Villin-GFP*, *Villin-RasV12*, or *APC^Min-Villin-RasV12* mice were injected with low-dose tamoxifen and were sacrificed 3 days later. Frozen sections were stained with DAPI (blue), anti-GFP antibody (green), and phalloidin (red). In **c**, **e**, **f**, intestinal organoids from *APC^Min-Villin-GFP*, *Villin-RasV12*, or *APC^Min-Villin-RasV12* mice were treated with 100 nM (**c**, **e**) or 1 µM tamoxifen (**f**) and cultured for the indicated times (**d**) or 24 h (**e**, **f**). Ba and Ap stand for the basal and apical sides, respectively (**a**, **c**, **f**). White squares in the left images are magnified in the corresponding right panels (**a**, **c**, **f**). White arrowheads depict the apically extruded cells whereas white arrows indicate the basally extruded cells, respectively (**a**, **c**, **e**). In (**f**), An arrow indicates a single transformed cell that is basally extruded (left magnified image) while the clustered transformed cells that are not extruded are depicted by an arrowhead (right magnified image). Scale bars, 50 µm (**a**, **c**), 100 µm (**e**), and 10 µm (**f**). **b**, **d**, **g** Quantification of apical and basal extrusion of transformed cells in small intestine (**b**) or intestinal organoids (**d**, **g**). 'Not extruded' indicates cells remaining within epithelium. 'Apical extruding' indicates apically extruding cells, with their nucleus apically shifted, that are still attached to the basement membrane. 'Apical extruded' indicates cells completely detached from the basement membrane and 'Basal extrusion' indicates cells delaminating into the basal lamina. Data are mean ± s.e.m. *n* = three independent experiments. *P = 0.0054 for basally extruded cells between RasV12- and APC^Min/RasV12-expressing cells (**b**), unpaired two-tailed *t*-test. In **g**, the clusters are categorized by the number of APC^Min/RasV12 cells present in each cluster. Data are mean ± s.e.m. *n* = three independent experiments. *P = 0.0037; **P = 0.0007 for cells with extrusion, unpaired two-tailed *t*-test.

not express synaptophysin or chromogranin, markers for enteroendocrine cells, confirming that they were not derived from enteroendocrine cells (Supplementary Fig. 5b). These results indicate that Wnt-induced perturbation of cell competition leads to the generation of highly invasive, unique carcinoma cells derived from terminally differentiated cells, and follow a distinct histopathogenesis pathway from the well-known adenoma-carcinoma sequence. Importantly, APC^Min/RasV12-positive adenomatous structures, where GFP-positive transformants occupied positions ranging from crypts to differentiated tips of villi, were also observed, albeit rarely, plausibly representing stem cell-hit lesions. Of particular interest, APC^Min/RasV12-transformed cells actively infiltrated stromal vessels (Fig. 2c), and three-dimensional imaging of the small intestine by whole-mount staining revealed their preferential invasion of lymphatic vessels (Fig. 2d). Moreover, *APC^Min-Villin-RasV12* mice with lymphatic invasion invariably exhibited metastasis of tumor cells to mesenteric lymph nodes (Fig. 2e). In summary, dysregulation of cell competition caused by Wnt activation can be exploited by RasV12-transformed differentiated cells to manifest diffuse invasion, resulting in the production of highly invasive carcinoma cells, which follows a distinct fate from the adenoma–carcinoma sequence.

## MMP21 regulates non-cell autonomous basal extrusion of APC^Min/RasV12 cells

With the aim of understanding the molecular mechanism underlying the non-cell autonomous basal delamination of transformed cells, we established Madin-Darby canine kidney (MDCK) or MDCK-pTR GFP-RasV12 cells (tetracycline-inducible RasV12-expressing cells) stably expressing a mCherry-conjugated β-catenin mutant (β-cat ΔN) to mimic Wnt activation in vitro (Fig. 3a). This mutant lacks the N-terminal 131 amino acids and fails to undergo GSK3β-mediated degradation, thereby resulting in constitutive activation of Wnt signaling[33]. We confirmed that MDCK cells expressing a β-cat ΔN mutant (referred to as β-cat ΔN cells) and MDCK-pTR GFP-RasV12 cells expressing the same mutant (referred to as β-cat ΔN/RasV12 cells) exhibit comparable Wnt activation by TOP FLASH reporter assay (Fig. 3b). When β-cat ΔN/RasV12-transformed cells were co-cultured with β-cat ΔN cells at a ratio of 1:50, a sizeable fraction of β-cat ΔN/RasV12 cells underwent basal extrusion into the collagen matrix over time, while the number of apically extruded cells or dead cells slightly increased compared to β-cat ΔN/RasV12 cells cultured alone (Fig. 3c, d). In addition, we found that β-cat ΔN/RasV12-transformed cells surrounded by themselves remained within epithelia, indicating that the presence of surrounding β-cat ΔN cells greatly accelerated basal extrusion of β-cat ΔN/RasV12 cells. Thus, we successfully phenocopied the salient feature of cell density-dependent basal extrusion of APC^Min/RasV12 cells in vitro.

We next wondered whether Wnt-activated cells would be eliminated by cell competition upon the emergence of Wnt-transformed cells within normal epithelia. To address this question, we established MDCK cells stably expressing doxycycline-inducible β-cat ΔN cells (Supplementary Fig. 6a). After 16 h of doxycycline treatment, Wnt

signaling was profoundly activated (Supplementary Fig. 6b). However, we did not observe any apparent "loser" phenotypes such as cell extrusion or cell death when β-cat ΔN cells were surrounded by normal MDCK cells (Supplementary Fig. 6c, d). This indicates that Wnt activation alone does not confer loser status, at least under these experimental conditions. It is worth noting that in other systems, such as zebrafish or hair follicles of mice, cells with activated Wnt signaling have been reported to be expelled from tissues via cell competition[34,35], suggesting that the removal of Wnt mutants by cell competition may be context-dependent.

To uncover the underlying molecular mechanism whereby Wnt activation potentiates basal extrusion of β-cat ΔN/RasV12-transformed cells, we conducted microarray analyses to search for differentially expressed genes in β-cat ΔN/RasV12 cells when co-cultured with β-cat ΔN cells (the data were deposited into the Gene Expression Omnibus (GEO) under accession number GSE217830). Among the top upregulated genes, we identified MMP21, a member of the matrix metalloproteases (MMPs) superfamily (Fig. 3e, f). MMPs play crucial roles in tumor progression by degrading and remodeling extracellular matrix (ECM), facilitating the invasion of malignant cells into stomal tissues[36]. This led us to assess expression levels of other MMP molecules by q-PCR analysis, which revealed that MMP21 was by far the most abundant MMP expressed in β-cat ΔN/RasV12 cells co-cultured with β-cat ΔN cells (Fig. 3g). These findings suggest that MMP21 may have a unique function among MMPs and could represent a particular target for the treatment of diffuse-type carcinomas. Immunofluorescence analysis, however, did not show overt alteration in endogenous MMP21 expression. MMP21 is synthesized as a 62-kDa proprotein, which is then activated through cleavage of the prodomain by furin or related proteases, resulting in the secretion of the active 49-kDa protease into the extracellular milieu[37]. For this reason, we surmised that synthesized MMP21 is rapidly secreted out of cells, making it difficult to observe intracellular MMP21. To circumvent this difficulty, we hampered the transport machinery by treating cells with brefeldin A (BFA). This manipulation led to the accumulation of MMP21 in β-cat ΔN/RasV12 cells when surrounded by β-cat ΔN cells, whereas MMP21 levels in β-cat ΔN/RasV12 cells cultured alone in the presence of BFA elevated to a lesser extent (Fig. 3h, i). Human *MMP21* is the most recently cloned MMP gene[38], and its physiological function, including substrate specificity, is not fully understood. Based on its amino acid sequence, MMP21 cannot be classified as a collagenase, gelatinase, stromelysin, matrilysin, or membrane-type MMP[38]. To set out to characterize the proteolytic capabilities of MMP21, we produced a recombinant protein of the catalytic domain of MMP21 using *E. coli* since the latent MMP21 protein is catalytically inert (Supplementary Fig. 7a). This recombinant protein was incubated with various ECMs, which are principal constituents of connective tissues. The catalytic domain of MMP21 extensively degraded collagen type IV and digested collagen type I and fibronectin, generating both small and large fragments. In contrast, laminin was resistant to hydrolysis, suggesting that MMP21 is a competent $Zn^{2+}$-dependent endoproteinase with unique specificity for

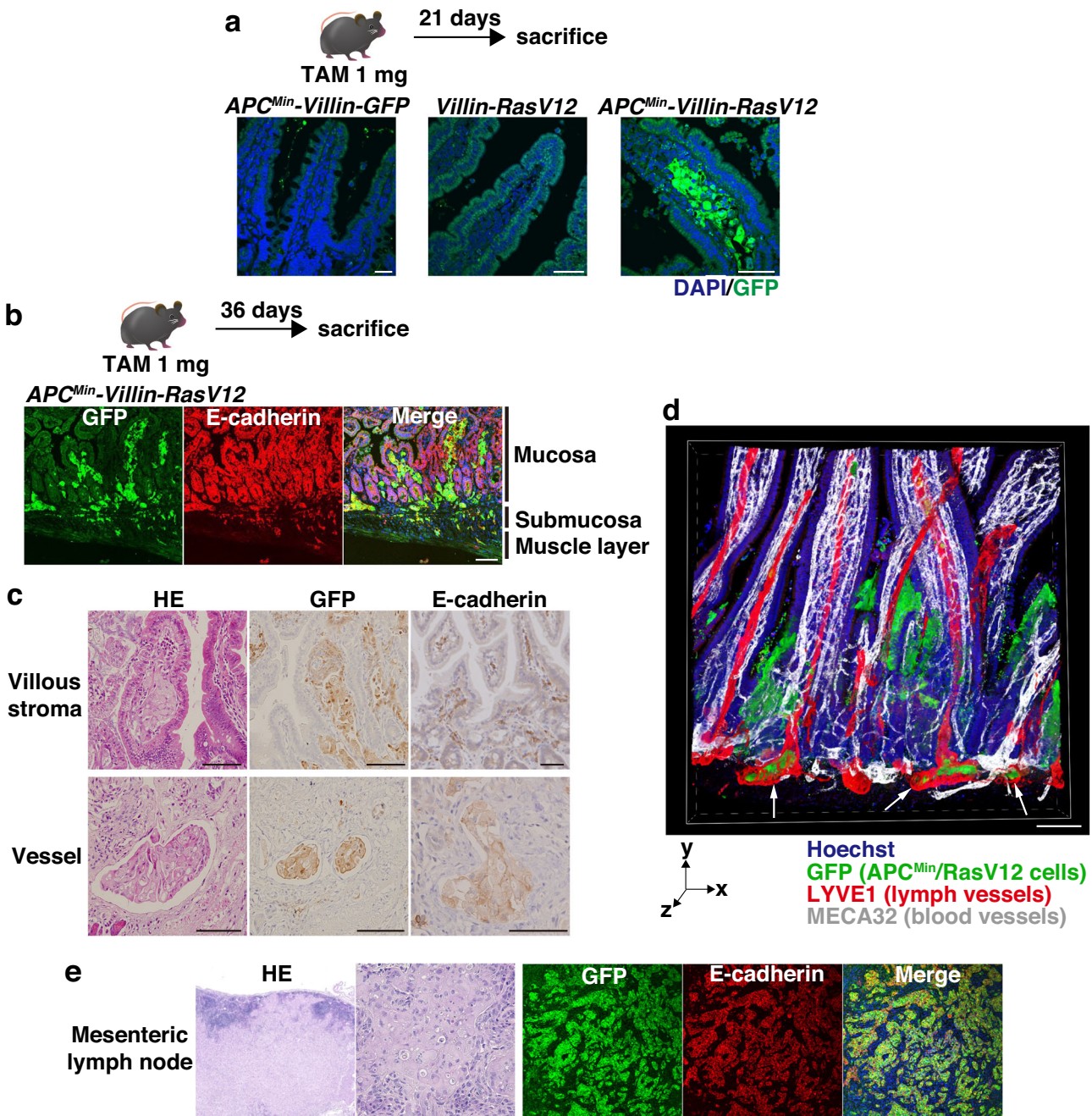

**Fig. 2 | APC^Min^/RasV12-transformed cells develop into diffusive carcinomas and invade lymphatic vessels. a, b** Confocal microscopic images of intestinal villi from *APC^Min^-Villin-GFP*, Villin-RasV12, or *APC^Min^-Villin-RasV12* mice. *APC^Min^-Villin-GFP*, *Villin-RasV12*, or *APC^Min^-Villin-RasV12* mice were injected with 1 mg tamoxifen and were sacrificed 21 days (**a**) or 36 days (**b**) later. Frozen sections were stained with DAPI (blue), anti-GFP antibody (green), and E-cadherin (red, **b**). **c** HE and immunostaining of intestinal villi from *APC^Min^-Villin-RasV12* mice bearing carcinomas 42 days after RasV12 expression. Paraffin-embedded sections were processed for HE staining or were stained with GFP or E-cadherin antibody (brown). **d** 3D image of whole mount-stained intestinal villi from *APC^Min^-Villin-RasV12* mice bearing carcinomas. A fixed sample was stained with Hoechst 33342 (blue), GFP (green), LYVE1 (red), and MECA32 (white). White arrows depict APC^Min^/RasV12 cells that invade lymph vessels. **e** HE and immunostaining of mesenteric lymph nodes of *APC^Min^-Villin-RasV12* mice. Paraffin-embedded sections were processed for HE staining or were stained with DAPI (blue), GFP (green), and E-cadherin antibody (red). Scale bars, 50 μm (**a**, **c**, **e**) and 100 μm (**b**, **d**).

certain substrates (Supplementary Fig. 7b). To examine the functional involvement of MMP21 in basal extrusion of transformants, we generated β-cat ΔN/RasV12 cells stably expressing MMP21 short-hairpin RNA (shRNA) (Fig. 3j). *MMP21*-knockdown profoundly prevented the basal extrusion of β-cat ΔN/RasV12 cells (Fig. 3k, l), highlighting an active role of MMP21 in the non-cell autonomous basal extrusion of transformed cells. To bolster evidence for the requirement of MMP21 in this process, we investigated the effect of a Pan-MMP inhibitor,

GM6001, and found that the frequency of basal extrusion was compromised to the same extent as *MMP21*-knockdown (Supplementary Fig. 7c, d).

We further investigated the functional relevance of MMP21 in the basal invasion of APC^Min^/RasV12 cells in vivo. To begin with, we examined MMP21 expression in intestinal epithelia and observed profound upregulation of MMP21 in RasV12-expressing cells that were produced mosaically within the APC^Min^-mutated epithelia, but not in

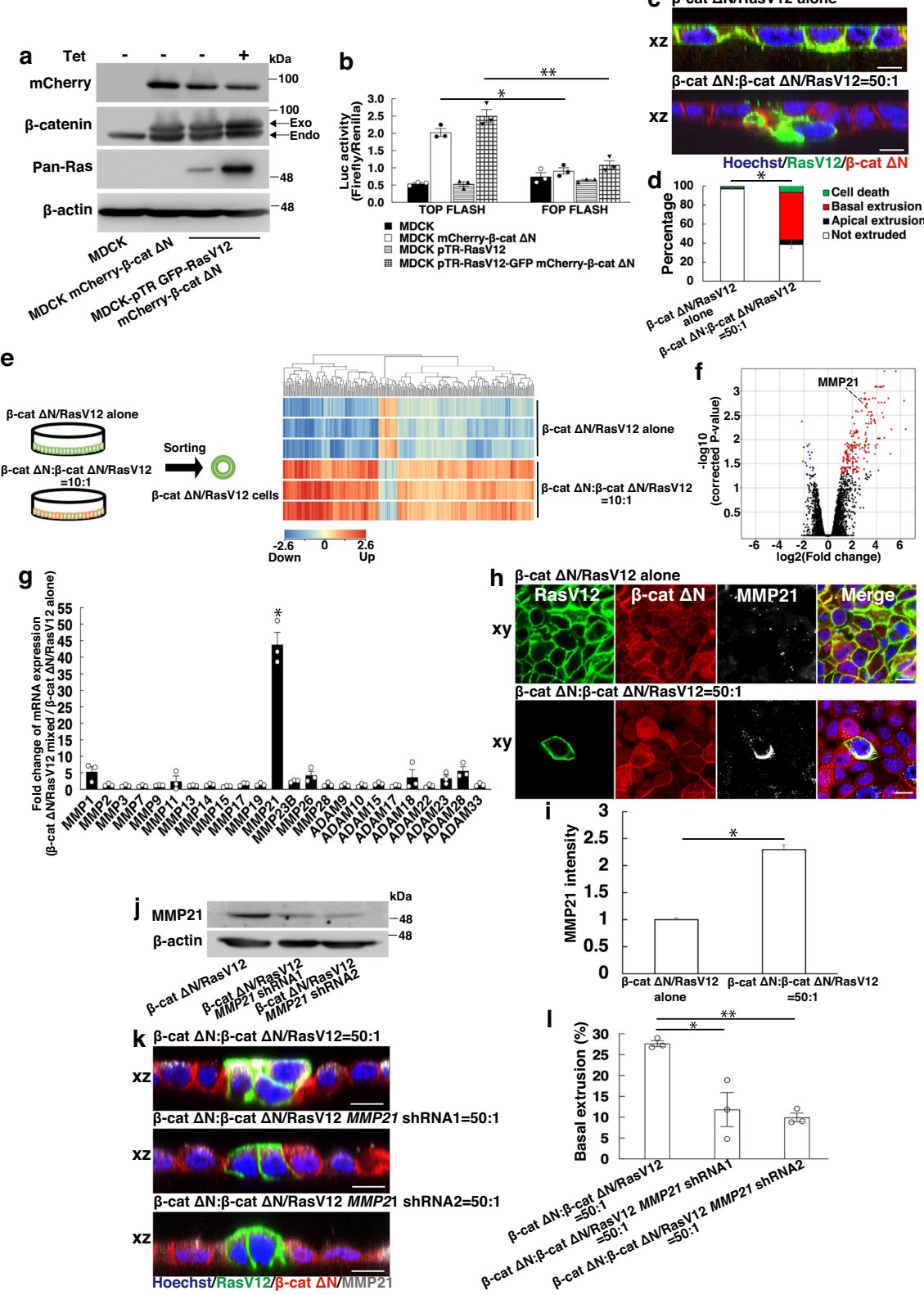

GFP-expressing cells nor in single RasV12-transformed cells (Fig. 4a). To investigate the cell density-dependent effect on MMP21 expression, intestinal organoids derived from *APC^Min-Villin-RasV12* mice were treated with either low-dose or high-dose tamoxifen. Mosaic expression of RasV12 mutants under the APC^Min-background induced elevated expression of MMP21, but not in APC^Min/RasV12 cells surrounded by one another, nor in APC^Min/GFP cells or RasV12 cells (Fig. 4b),

recapitulating the cell density-dependent upregulation of MMP21 in APC^Min/RasV12 cells. It is worth noting that MMP21 was upregulated not only in basally extruded cells but also in cells remaining within the epithelia or even apically extruding cells. This suggests that the elevated expression of MMP21 alone is not sufficient to redirect transformed cells to undergo basal delamination, but rather it promotes this process as one of the priming factors. To examine the functional

**Fig. 3 | MMP21 plays a crucial role in basal extrusion of β-cat ΔN/RasV12 cells surrounded by β-cat ΔN cells. a, b** Establishment of MDCK mCherry-β-cateninΔ131 and MDCK-pTR GFP-RasV12 mCherry-β-cateninΔ131 cells. **a** Constitutive expression of mCherry-β-cateninΔ131 and tetracycline-induced expression of RasV12. **b** TOP FLASH reporter assay. Indicated cell lines were co-transfected with TOP FLASH or FOP FLASH and Renilla expressing vector, and subjected to measurement of luciferase activity. Data are mean ± s.e.m. *P = 0.0015; **P = 0.0032, unpaired two-tailed *t*-test; *n* = three independent experiments. **c** Immunofluorescence images of β-cat ΔN/RasV12 cells cultured alone or surrounded by β-cat ΔN cells. **d** Quantification of cell fates of β-cat ΔN/RasV12 cells cultured alone or surrounded by β-cat ΔN cells. Data are mean ± s.e.m. *n* = three independent experiments. *P = 0.0190 for basal extrusion between two conditions, unpaired two-tailed *t*-test. **e** Experimental design and heatmap for differentially expressed genes between β-cat ΔN/RasV12 cells cultured alone and β-cat ΔN/RasV12 cells co-cultured with β-cat ΔN cells. **f** Volcano plot showing the distribution of fold changes and corrected *P*-values calculated by unpaired two-tailed *t*-test in β-cat ΔN/RasV12 cells mixed with β-cat ΔN cells compared with a single culture. The red or blue dots represent upregulated or downregulated genes with statistical significance, respectively. The upregulated and downregulated genes are shown in Supplementary Data 1. **g** q-PCR analysis of MMP-related metalloproteases. β-cat ΔN/RasV12 cells were co-cultured with β-cat ΔN cells or cultured alone, and subjected to q-PCR analysis. Values are shown as fold change relative to β-cat ΔN/RasV12 cells cultured alone. Data are mean ± s.e.m. *n* = three independent experiments. *P = 0.0221, unpaired two-tailed *t*-test. **h** Immunofluorescence images of MMP21. β-cat ΔN/RasV12 cells were cultured alone or co-cultured with β-cat ΔN cells in the presence of 2.5 μg ml⁻¹ BFA. Cells were stained with Hoechst 33342 (blue) and anti-MMP21 antibody (white). **i** Quantification of fluorescence intensity of MMP21. Values are expressed as a ratio relative to β-cat ΔN/RasV12 cells cultured alone. Data are mean ± s.e.m. *P < 0.001, unpaired two-tailed *t*-test; *n* = 286 and 284 cells pooled from three independent experiments. **j** Establishment of MDCK-pTR GFP-RasV12 mCherry-β-cateninΔ131 cells stably expressing *MMP21* shRNA1 or *MMP21* shRNA2. Knockdown of *MMP21* was confirmed by western blotting. **k** Immunofluorescence images of β-cat ΔN/RasV12 cells, β-cat ΔN/RasV12 cells expressing *MMP21* shRNA1 or *MMP21* shRNA2 surrounded by β-cat ΔN cells. Scale bars, 10 μm (**c**, **h**, **k**). **l** Quantification of basal extrusion. Data are mean ± s.e.m. *n* = three independent experiments. *P = 0.0186; **P = 0.0001, unpaired two-tailed *t*-test.

---

significance of MMP21 in basal delamination from normal mucosa, we utilized MMP21-deficient mice (Fig. 4c–f). Although the majority of the progeny (93.9%; *n* = 132 mice) died immediately after birth, likely due to congenital heart disease caused by MMP21 abrogation[39,40], we analyzed the *APC^Min^-Villin-RasV12-MMP21 KO* mice that were born healthy, grew normally, and did not exhibit apparent diseases. Consequently, *MMP21*-deficient APC^Min^/RasV12 cells were incompetent to drive the basal extrusion of transformed cells (Fig. 4g, h). These results collectively demonstrate that MMP21 is one of the molecules that govern the diffuse invasion of transformed cells.

## NF-κB signaling directly elevates MMP21 expression

To identify the upstream regulator(s) responsible for the upregulation of MMP21 expression, we initially examined the involvement of the Notch and TGF-β pathways as potential regulators of MMP21[41,42]. However, no alterations in activities of these pathways were evident by examining the transcriptional signature of β-cat ΔN/RasV12 cells co-cultured with β-cat ΔN cells, armed with the microarray datasets (Fig. 3e). Subsequently, we conducted gene-set enrichment analyses (GSEA) to gain a comprehensive overview of differentially expressed genes, and found that the production of IL-6 and chemokines substantially increased (Fig. 5a). This result is suggestive of activation of the NF-κB signaling pathway[43], we thereby conducted a reporter assay using a plasmid with luciferase expression under the control of tandem repeats of the NF-κB transcriptional response element. Consequently, we validated the non-cell-autonomous activation of NF-κB signaling (Fig. 5b). Notably, we observed negligible activation of Wnt signaling (Supplementary Fig. 8), indicating that further enhancement of Wnt signaling does not occur in β-cat ΔN/RasV12 cells. To explore whether the upregulation of MMP21 and NF-κB activation induced in β-cat ΔN/RasV12 cells are specific to a Wnt-activated background, we performed microarray analysis to profile RasV12-single transformed cells co-cultured with normal cells, alongside RasV12 cells cultured alone (GEO accession Number: GSE236658). As a result, we did not observe a significant elevation of MMP21, and NF-κB activation was found to be negligible. These findings suggest that a Wnt-activated environment is necessary for these alterations to occur. To investigate the functional relevance of the NF-κB pathway in MMP21-mediated diffuse invasion of transformed cells, we treated the cells with BAY 11-7082, an inhibitor of NF-κB that targets IκB kinase (IKK). Consequently, BAY 11-7082 substantially diminished the cell density-dependent upregulation of MMP21, suggesting that NF-κB signaling acts as an upstream regulator of MMP21 expression (Fig. 5c, d). We next queried whether the NF-κB complex directly regulates MMP21 transcription. To this end, we performed chromatin immunoprecipitation (ChIP) assay followed by q-PCR using primers encompassing the promoter region of the *MMP21*

locus, which contains the predicted p65 binding sequence according to the JASPAR2014 program (Fig. 5e). The ChIP q-PCR experiments demonstrated that activation of NF-κB signaling by TNF-α treatment significantly increased the association of p65 with the promoter region of *MMP21*, indicating direct regulation of MMP21 expression by the NF-κB signaling pathway (Fig. 5e). Furthermore, inhibition of NF-κB signaling profoundly suppressed the basal extrusion of β-cat ΔN/RasV12 cells co-cultured with β-cat ΔN cells, and adversely enhanced their apical elimination in a dose-dependent manner (Fig. 5f, g). These results indicate that NF-κB signaling is crucial in determining the direction of cell extrusion, and its activation redirects transformed cells toward basal delamination, with MMP21 being one of the downstream targets.

To evaluate the activity of NF-κB signaling in vivo, we examined the expression of p65 and observed a profound accumulation of nuclear p65 in APC^Min^/RasV12 cells within the intestinal epithelia, which was otherwise unaffected in APC^Min^/GFP- or single RasV12-mutated cells (Fig. 5h). Furthermore, p65 elevation occurs in a cell density-dependent fashion, as observed in intestinal organoids where APC^Min^/RasV12 cells surrounded by APC^Min^ cells exhibited increased nuclear p65 levels whereas p65 intensity in APC^Min^/RasV12 mutants surrounded by themselves was no higher than that in RasV12 or APC^Min^/GFP cells (Fig. 5i). To investigate the impact of NF-κB signaling inhibition on MMP21 expression and the basal extrusion of APC^Min^/RasV12 cells, we treated intestinal organoids with BAY 11-7082. The treatment drastically diminished the cell density-dependent upregulation of MMP21 and counteracted the basal extrusion of APC^Min^/RasV12 cells (Fig. 5j–l). Additionally, we challenged *APC^Min^-Villin-RasV12* mice with SN50, a cell-permeable NF-κB inhibitory peptide, and observed a significant decrease in the number of basally extruded cells compared to the group administered with PBS (Fig. 5m, n). These in vivo findings support our conclusion that NF-κB signaling positively regulates basal invasion of transformed cells through MMP21 upregulation.

To elucidate the mechanisms underlying NF-κB activation, we re-analyzed the microarray data, where the expression profile of β-cat ΔN/RasV12 cells co-cultured with β-cat ΔN cells was compared to that of β-cat ΔN/RasV12 cells cultured alone (Fig. 3e). Metascape analysis revealed a marked enhancement of the innate immune response including interferon production (Supplementary Fig. 9a). Notably, the retinoic acid-inducible gene-I (RIG-I) pathway emerged as one of the top-activated pathways. Additionally, Toll-like Receptor (TLR) signaling was found to be activated, with TLR3 showing the highest upregulation among the TLR family. This led us to further investigate the expression of RIG-I (also referred to as DDX58) and TLR3. Accordingly, we performed q-PCR analysis for each transcript and observed accelerated expression of both molecules in β-cat ΔN/RasV12 cells co-

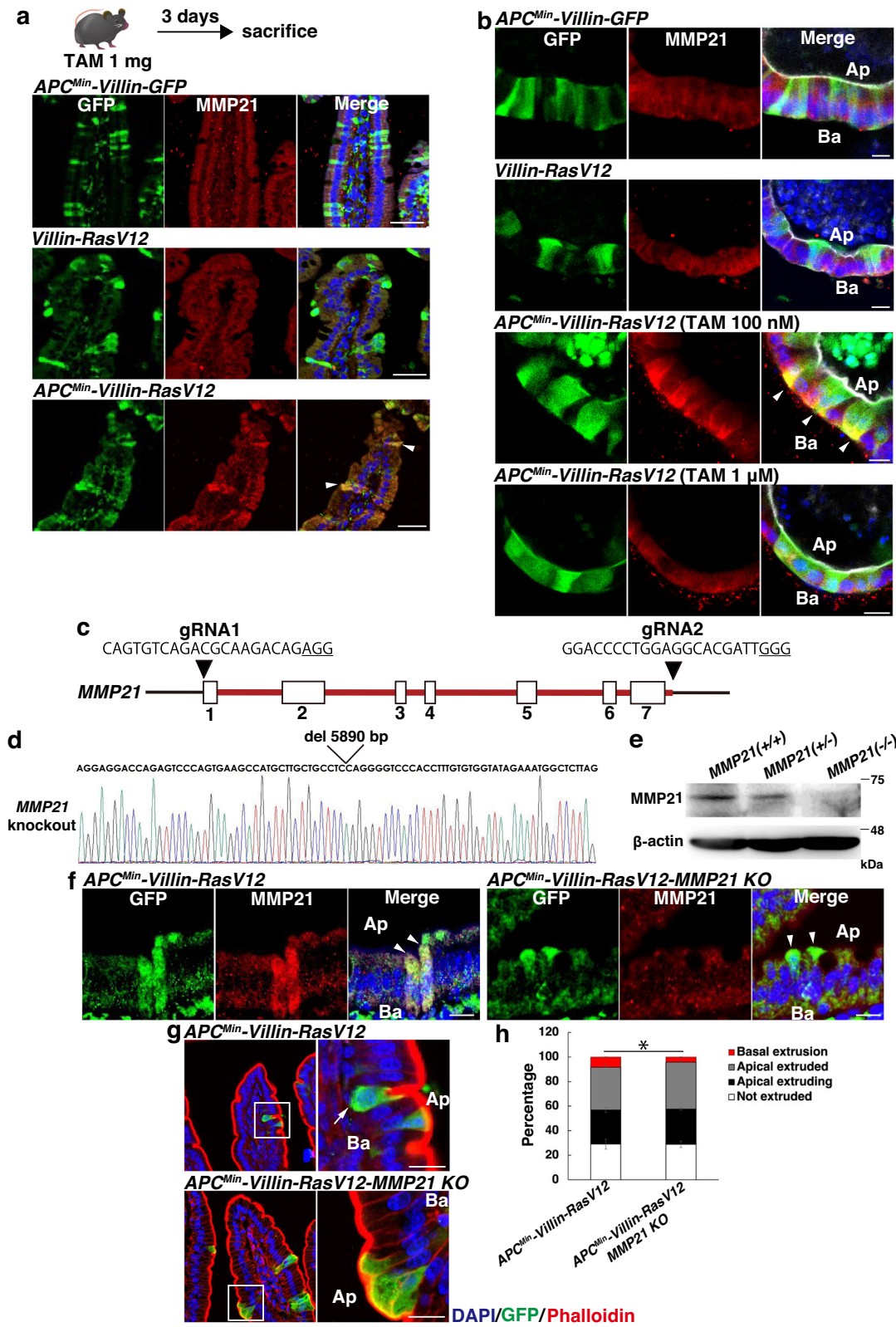

cultured with β-cat ΔN cells, surpassing their expression in β-cat ΔN/RasV12 cells in homotypic setting (Supplementary Fig. 9b). As RIG-I and TLR3 are well-known pattern recognition receptors to sense intracellular DNA/RNA and potentiate the innate immune response through NF-κB pathway[44,45], we examined the inhibitory effect of these pathways on the cell fates of transformed cells. As a result, inhibition of RIG-I or TLR3 by RIG012 or CU-CPT 4a, respectively, significantly suppressed non-cell autonomous NF-κB activation and MMP21 upregulation (Supplementary Fig. 9c–e). Moreover, the frequency of basal extrusion was reduced, while apical extrusion was adversely increased, recapitulating the effect of NF-κB inhibition (Supplementary Fig. 9f, g). In organoid cultures, treatment with RIG012 or CU-CPT 4a yielded comparable effects to those observed in vitro: suppression of NF-κB activation and MMP21 elevation, and reduction of basal extrusion

**Fig. 4 | MMP21 promotes the non-cell autonomous basal invasion of APC[Min]/RasV12-transformed cells in vivo. a, b** The non-cell autonomous upregulation of MMP21 in APC[Min]/RasV12 cells. **a** Immunofluorescence images of MMP21 in intestinal villi. *APC[Min]-Villin-GFP, Villin-RasV12,* or *APC[Min]-Villin-RasV12* mice were injected with tamoxifen and were sacrificed 3 days later. Paraffin-embedded sections were stained with DAPI (blue), anti-GFP antibody (green), and anti-MMP21 antibody (red). **b** Immunofluorescence images of MMP21 in intestinal organoids from *APC[Min]-Villin-GFP, Villin-RasV12,* or *APC[Min]-Villin-RasV12* mice at 24 h after tamoxifen treatment. Arrowheads indicate APC[Min]/RasV12-transformed cells with elevated MMP21 expression (**a, b**). **c–h** Effect of *MMP21*-knockout on basal extrusion of APC[Min]/RasV12 cells. **c** Strategy for the CRISPR-mediated *MMP21*-targeted mice. Physical maps of the murine *MMP21* gene locus and sgRNA sequences are shown. The red line indicates the targeted region. Protospacer adjacent motif (PAM) sequences are underlined. **d** Genomic sequence of *MMP21* locus from F1 generation *MMP21*-knockout mice. Note that 5890 bp covering Exon 1–Exon 7 of the *MMP21* gene is deleted. **e** Western blot analysis of protein lysates prepared from intestinal tissues of *MMP21(+/+), (+/−), (−/−)* genotypes. β-actin served as a loading control. **f** Immunofluorescence images of MMP21 in intestinal villi from *APC[Min]-Villin-RasV12* or *APC[Min]-Villin-RasV12-MMP21 KO* mice. Paraffin-embedded sections were stained with DAPI (blue), anti-GFP antibody (green), and anti-MMP21 antibody (red). Arrowheads indicate APC[Min]/RasV12-transformed cells. **g, h** Frequency of the basal extrusion of APC[Min]/RasV12 cells. **g** Immunofluorescence images of intestinal villi from *APC[Min]-Villin-RasV12* or *APC[Min]-Villin-RasV12-MMP21 KO* mice 3 days after tamoxifen injection. An arrow indicates a basally extruded cell. Ba and Ap stand for the basal and apical sides, respectively (**b, f, g**). Scale bars, 50 μm (**a**) and 10 μm (**b, f, g**). **h** Quantification of apical and basal extrusion of transformed cells. Data are mean ± s.e.m. *n* = three independent experiments. *P = 0.0004, unpaired two-tailed *t*-test.

(Supplementary Fig. 9h–j). Based on these findings, it is suggested that the competitive environment entails the RIG-I/TLR3-mediated innate immune response for Wnt-activated RasV12-transformed cells, priming them for basal infiltration through the NF-κB-MMP21 pathway (Supplementary Fig. 10).

## The NF-κB-MMP21 pathway is activated in early colorectal cancers in humans

Finally, we investigated whether the molecular mechanism underlying the onset of invasive carcinomas developed in *APC[Min]-Villin-RasV12* mice is involved in human colorectal cancer. However, the cancer histopathology of *APC[Min]-Villin-RasV12* mice does not mimic any specific pathological classes of regular human colorectal cancer, thereby, it is hard to obtain corresponding human clinical samples. Moreover, cell competition is the event occurring in the initial phase of carcinogenesis. We therefore collected 9 tissue specimens from patients diagnosed with early colorectal cancer who received endoscopic treatments. These samples were stained for MMP21, p65, β-catenin, or p-ERK. β-catenin and p-ERK were analyzed to assess the activity of Wnt signaling and MAPK signaling, respectively. Histological scores (H-scores) were calculated based on the percentages of positive cells showing, negative, weak, moderate, or strong staining intensity (Fig. 6a). Our analysis revealed higher expression of MMP21, nuclear β-catenin, p-ERK, and nuclear p65 compared to normal tissues (Fig. 6b). Furthermore, the H-score of MMP21 positively correlated with that of nuclear β-catenin, p-ERK or nuclear p65 in tumor tissues (Fig. 6c). In contrast, the H-scores of p-Akt and Cyclin E1, which are known to frequently increase in the relatively advanced stages of colorectal cancers[46,47], did not correlate with the H-score of MMP21, despite their higher expression in tumors (Fig. 6b, c). Collectively, these results suggest that the NF-κB–MMP21 pathway is bolstered in early colorectal cancers with activated Wnt and MAPK signaling, potentially playing a role in the early infiltration of malignant cells in humans.

## Discussion

There has been a surge of interest in cell competition for developing novel anti-cancer therapy. To achieve this goal, it is crucial to fully understand whether and how cell competition malfunctions during the onset of carcinogenesis. In this study, we demonstrated that preceding Wnt activation tips the balance in cell competition-induced cellular extrusion, which leads to adverse outcomes, illuminating an unanticipated outcome of cell competition.

Regarding the molecular mechanisms underlying APC[Min]-induced functional perturbation of cell competition, we carefully investigated potential cellular aberrations that could be attributed to the disoriented extrusion of RasV12-transformed cells. Previous studies have reported that spindle misorientation is associated with irregular extrusion in APC-mutated dividing crypt cells[48–50]. However, basally extruding APC[Min]/RasV12 cells observed in our study were terminally differentiated and non-proliferative enterocytes, suggesting that abnormal cell division is not responsible for basal extrusion. Also, APC is involved in the regulation of cellular polarity[51,52]. This led us to examine the localization of junctional molecules, but we did not observe any disturbances in either APC[Min]/RasV12 cells or β-cat ΔN/RasV12 cells. From a molecular standpoint, our findings indicate that activation of RIG-I and TLR3 signaling pathways potentiates NF-κB signaling in APC[Min]/RasV12 cells surrounded by APC[Min] cells. This leads to elevated MMP21 expression, which is crucial for basal delamination. Moreover, we discovered that NF-κB signaling acts as a key determinant to switch the predominant direction of extrusion, shifting it from apical to basal. These findings align with previous studies demonstrating the role of NF-κB signaling in chronic inflammation, where it serves as a master regulator connecting inflammation and cancer, promoting cancer cell invasion[53]. While MMP21 abrogation did not fully recapitulate the phenotype observed with NF-κB inhibition, it suggests that MMP21 acts as a downstream effector of NF-κB signaling in driving diffuse invasion. Further investigation is needed to clarify how the molecular landscape is remodeled to alter the cell fates of APC[Min]/RasV12 compound mutants, which could be harnessed for therapeutic applications.

MMPs comprise a family of topologically related zinc endoproteases that degrade components of the extracellular matrix, or cause the shedding of membrane-anchored proteins and are involved in cancer, arthritis, multiple sclerosis, and cardiovascular disease[54]. Despite sharing a conserved catalytic domain containing a zinc-binding motif (HEXXHXXGXXH), substrate specificity differs widely. In this study, we demonstrated that MMP21 exerts catalytic activity against collagen type I, type IV, and fibronectin. Although the upstream regulators for MMP21 expression were unclarified, our findings indicate that NF-κB signaling directly regulates the transcription of *MMP21*. However, the fact that inhibition of NF-κB signaling does not completely abolish the non-cell autonomous upregulation of MMP21 and expression of several MMPs is also under the control of NF-κB activity suggests the involvement of additional factors beyond NF-κB signaling in the regulation of MMP21 expression[55]. Previous studies have shown that increased expression of MMP21 is associated with poor clinical outcomes in several types of cancers[56]. Additionally, in this study, we observed significant upregulation of MMP21 in early human colorectal cancers, and its expression positively correlated with NF-κB signaling, Wnt signaling, and MAPK pathways. Thus, the present study is indicative of MMP21 as a potential therapeutic target for suppressing the expansion of early-phase tumors.

The adenoma-carcinoma sequence model for colorectal cancer is broadly acknowledged[57]. Nevertheless, our mouse model delineates a distinct mode of tumourigenesis, where malignant cells are diffusely generated within otherwise histologically normal tissues without the presence of adenoma components in proximity to tumors. What causes this difference in adenoma-carcinoma cancer or the histopathogenesis of *APC[Min]-Villin-RasV12* mice is of particular interest. One possible explanation is the different origins of cancer cells. In this

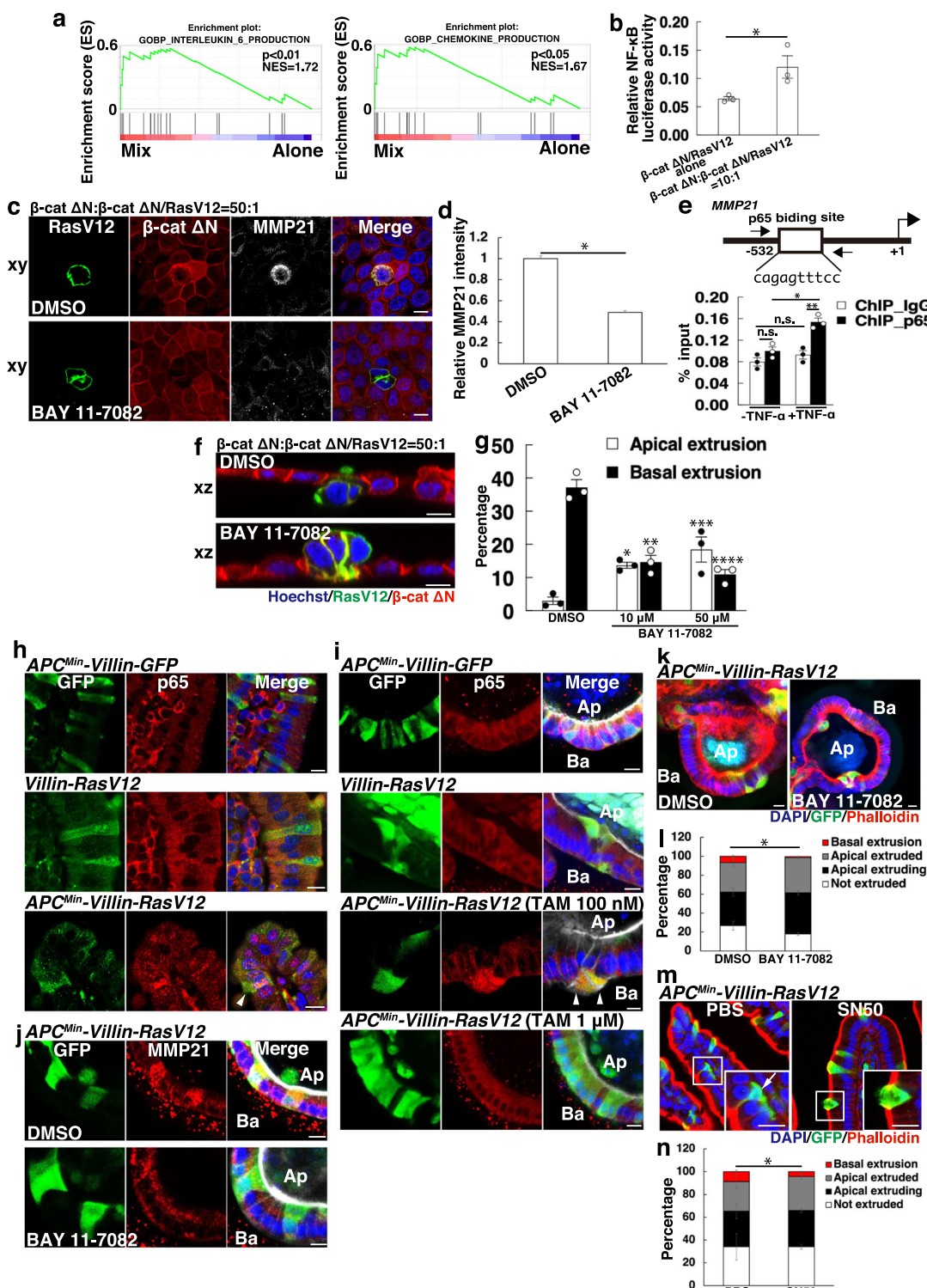

study, we engineered mice to induce RasV12 mutation in terminally differentiated enterocytes. However, it is well known that cancer arises from an incipient transformation event occurring primarily, but not exclusively, in the stem cell compartment. This is because stem cells in various organs are intrinsically vulnerable to genetic mutations due to their rapid cell division[58,59]. Hence, the current paradigm suggests that cancer arises from stem cells bearing sporadic mutation(s) that are inherited by all their progeny[60]. This renders a clonal advantage to transformed cells, leading to the development of adenomatous transition, followed by incidental production of invasive cells in later

stages. Interestingly, several recent studies have reported that APC-mutated crypt clones exhibit a competitive advantage over the surrounding normal crypts, causing accelerated expansion of APC-mutated clones[61–63]. This suggests that APC-mutated crypts, which contain intestinal stem cells, act as winners in "crypt competition". Although studies of human colorectal cancer originally put forward stem cells as the origin of the disease, several models have demonstrated that differentiated cells can also become principal contributors to tumor development[64–67]. Thus, it is tempting to speculate that differentiated cells with deleterious mutations cause different cancerous

**Fig. 5 | NF-κB signaling governs basal extrusion of transformed cells via MMP21 upregulation. a, b** Enhanced NF-κB activity in β-cat ΔN/RasV12 cells co-cultured with β-cat ΔN cells. **a** GSEA plot showing a significant correlation between the transcriptional profile of β-cat ΔN/RasV12 cells co-cultured with β-cat ΔN cells and IL-6 or chemokine production gene signatures. The *P*-values were calculated based on a two-sided permutation test. **b** NF-κB reporter assay. β-cat ΔN/RasV12 cells were co-transfected with NF-κB luciferase reporter and Renilla expressing vector and were co-cultured with β-cat ΔN cells or cultured alone. Cells were then subjected to measurement of luciferase activity. Data are mean ± s.e.m. *n* = three independent experiments. *P = 0.0498, unpaired two-tailed *t*-test. **c–g** Effect of BAY 11-7082 on MMP21 expression and cell fates of β-cat ΔN/RasV12 cells. **c** Immunofluorescence images of MMP21 in the absence or presence of BAY 11-7082. β-cat ΔN/RasV12 cells were mixed with β-cat ΔN cells in the presence of BFA and were treated with DMSO or BAY 11-7082. Cells were stained with Hoechst 33342 (blue) and anti-MMP21 antibody (white). **d** Quantification of fluorescence intensity of MMP21. Values are expressed as a ratio relative to DMSO treatment. Data are mean ± s.e.m. *P < 0.001, unpaired two-tailed *t*-test; *n* = 418 and 420 cells pooled from three independent experiments. **e** p65 ChIP assay. An illustration of the p65 binding site within dog *MMP21* promoter region is shown in the upper picture. Effect of TNF-α on the recruitment of p65 NF-κB to *MMP21* promoter region is shown (lower panel). Data are mean ± s.e.m. *n* = three independent experiments. *P = 0.0069; **P = 0.0046, unpaired two-tailed *t*-test. **f** Immunofluorescence images of β-cat ΔN/RasV12 cells surrounded by β-cat ΔN cells in the absence or presence of BAY 11-7082. **g** Quantification of apical and basal extrusion in the absence or presence of BAY 11-

7082. Data are mean ± s.e.m. *n* = three independent experiments. *P = 0.0018; **P = 0.0019; ***P = 0.0176; ****P = 0.0007, unpaired two-tailed *t*-test. **h, i** Activation of NF-κB signaling in $APC^{Min}$/RasV12 cells. **h** Immunofluorescence images of p65 in intestinal villi. $APC^{Min}$-*Villin-GFP, Villin-RasV12*, or $APC^{Min}$-*Villin-RasV12* mice were injected with tamoxifen and were sacrificed 3 days later. Paraffin-embedded sections were stained with DAPI (blue), anti-GFP antibody (green) and anti-p65 antibody (red). **i** Immunofluorescence images of p65 in intestinal organoids from indicated mice. Arrowheads indicate $APC^{Min}$/RasV12-transformed cells with accumulated p65 in the nucleus (**h**, **i**). **j–n** Effect of NF-κB inhibition on MMP21 expression and cell fates. (**j**) Immunofluorescence images of MMP21 in intestinal organoids from $APC^{Min}$-*Villin-RasV12* mice in the absence or presence of BAY 11-7082. **k** Immunofluorescence images of intestinal organoids treated with 100 nM tamoxifen and cultured for 24 h in the absence or presence of BAY 11-7082. Ba and Ap stand for basal and apical sides, respectively (**i–k**). **l** Quantification of the effect of BAY 11-7082 on apical and basal extrusion ex vivo. Data are mean ± s.e.m. *n* = four independent experiments. *P = 0.0097 for basally extruded cells between DMSO- and BAY 11-7082-treated group, unpaired two-tailed *t*-test. **m** Immunofluorescence images of intestinal villi from $APC^{Min}$-*Villin-RasV12* mice administered with PBS or SN50. Magnified images depict basal (an arrow) or apical extrusion of transformed cells from PBS- or SN50-administered mice, respectively. Scale bars, 10 μm (**c**, **f**, **h–k**, **m**). **n** Quantification of extrusion of $APC^{Min}$/RasV12-transformed cells in the absence of the presence of SN50. Data are mean ± s.e.m. *n* = three independent experiments. *P = 0.0486 for basally extruded cells between PBS- and SN50-administered group, unpaired two-tailed *t*-test.

phenotypes, such as diffuse-type cancer, albeit rarely, due to the low incidence of somatic mutations in these cells. Supporting this notion, it has been estimated that patients with de novo diffuse-type colorectal cancer are quite rare or underestimated since it is barely detectable microscopically[68]. Furthermore, diffuse-type cancers generated in the digestive tract are difficult to treat, due to scarcely noticeable early symptoms and highly invasive characteristics of cancer cells. Therefore, it is of clinical significance to delineate the nature of the disease's pathogenesis, and there is a need for sensitive, novel treatments. Whether the NF-κB-MMP21 pathway is generally relevant to other diffuse-type cancers demands future investigation.

In conclusion, our findings demonstrate that Wnt activation disrupts cell competition, and confers invasive properties on transformed cells to escape primary epithelial sites. Previous studies have also shown that the fate of transformed cells upon cell competition can differ, depending on the preceding mutation background[69,70]. This study further brings forth the prospect that cell competition constrains the order of appearance of mutations during tumor development, highlighting a link between cell competition and carcinogenesis.

## Methods

### Antibodies and materials

The following antibodies were used in this study: chicken anti-GFP (ab13970), rabbit anti-Ki-67 (ab16667), rabbit anti-DCLK1 (ab31704), rabbit anti-LYVE1 (ab14917), mouse anti-chromogranin A (ab80787) and rabbit anti-p65 (ab16502) antibodies from Abcam, mouse anti-E-cadherin (clone 36;610181), mouse anti-β-catenin (clone 14;610153) and mouse anti-Villin (clone 12;610358) antibodies from BD Transduction, mouse anti-β-actin (clone AC-74;A2228), mouse anti-Pan-Ras (clone Ras10;OP40) and rabbit anti-laminin (L9393) antibodies from Sigma, mouse anti-mCherry antibody (632543) from Clontech, rabbit anti-Olfm4 (39141), rabbit anti-p-ERK (9101), rabbit anti-NF-κB (8242), rabbit anti-IgG (2729), rabbit anti-p-Akt (4060) and mouse anti-Cyclin E1 (4129) antibodies from Cell Signaling, rat anti-MECA32 (120502) antibody from BioLegend, mouse anti-CDX2 (MU392A-UC) antibody from BioGenex, rabbit anti-Sox9 (AB5535) antibody from Millipore and mouse anti-synaptophysin antibody (413831) from Nichirei Biosciences Inc. As for anti-MMP21 antibodies, rabbit anti-MMP21 (TA322032) antibody from ORIGENE was used for western blotting of MDCK lysate whereas rabbit anti-MMP21 (55289-1) antibody from Proteintech was used for mice tissues, and rabbit anti-MMP21 (PA1-

25234) antibody from Life Technologies was used for immunostaining. Alexa-Fluor-568- and -647-conjugated phalloidin (Life Technologies) were used at 1.0 U ml⁻¹. Alexa-Fluor-568- and -647-conjugated secondary antibodies were from Life Technologies, Alexa-Fluor-488-conjugated anti-chicken IgY antibody was from Abcam and Alexa-Fluor-647-conjugated anti-rat IgG was from Jackson immunoResearch. Tetramethylrhodamine-conjugated WGA was purchased from Life Technologies. Hoechst 33342 (Life Technologies) was used at a dilution of 1:2,000. The following inhibitors, BAY 11-7082 from Tokyo Chemical Industry, SN50 from Cayman Chemical, GM6001 from Calbiochem, Indomethacin from Sigma, RIG012 from Axon Medchem, and CU-CPT 4a from MedChemExpress were used.

### Mice

All animal experiments were conducted under the guidelines of the Animal Care Committee of Tokyo University of Science. The animal protocols were reviewed and approved by the Tokyo University of Science Animal Care Committee (Approval number: S21027). The maximal allowable tumor burden was a 20% reduction in mouse body weight, and this threshold was not exceeded in this study. We used 6–10 weeks old C57BL/6 mice of either sex. All mice were maintained in a specific pathogen-free environment, with a gamma-ray sterilized normal diet, acidified tap water (0.002 N HCl, pH 2.5), and autoclaved wooden chip bedding within environmentally controlled clean rooms. The animal facility maintained a 12 h/12 h light/dark cycle at 23–24 °C and a humidity level between 40–50% for housing the mice. *Villin-Cre^{ERT2}* mice[28] were crossed with *DNMT1-CAG-loxP-STOP-loxP-*HRas^{V12}-IRES-eGFP mice[17] or CAG-*loxP*-STOP-*loxP*-eGFP mice[71] and were further mated with $APC^{Min}$ mice[27] to create *Villin-RasV12, APC^{Min}-Villin-RasV12* or $APC^{Min}$-*Villin-GFP* mice, respectively. Mice heterozygous for each transgene were used for experiments. We obtained the *MMP21-null* mice by CRISPR/Cas9-mediated genome engineering from Cyagen Biosciences. The 5636 bp ranging across Exon 1–7 of *MMP21* gene located on chromosome 7 was selected as a target site (Fig. 4c). Cas9 and gRNA were co-injected into fertilized eggs for *MMP21*-knockout mice production. For PCR genotyping of mice, the following primers were used: 5′-CAAGCCTGGCTCGACGGCC-3′ and 5′-CGCGAACATCTT CAGGTTCT-3′ for the *Villin-Cre^{ERT2}* mice, 5′-CACTGTGGAATCTCGGC AGG-3′ and 5′-GCAATATGGTGGAAAATAAC-3′ for the *DNMT1-*CA G-*loxP*-STOP-*loxP*-HRas^{V12}-IRES-eGFP mice, 5′-CAGTCAGTTGCTCAATG TACC-3′ and 5′-ACTGGTGAAACTCACCCA-3′ for the CAG-*loxP*-STOP-

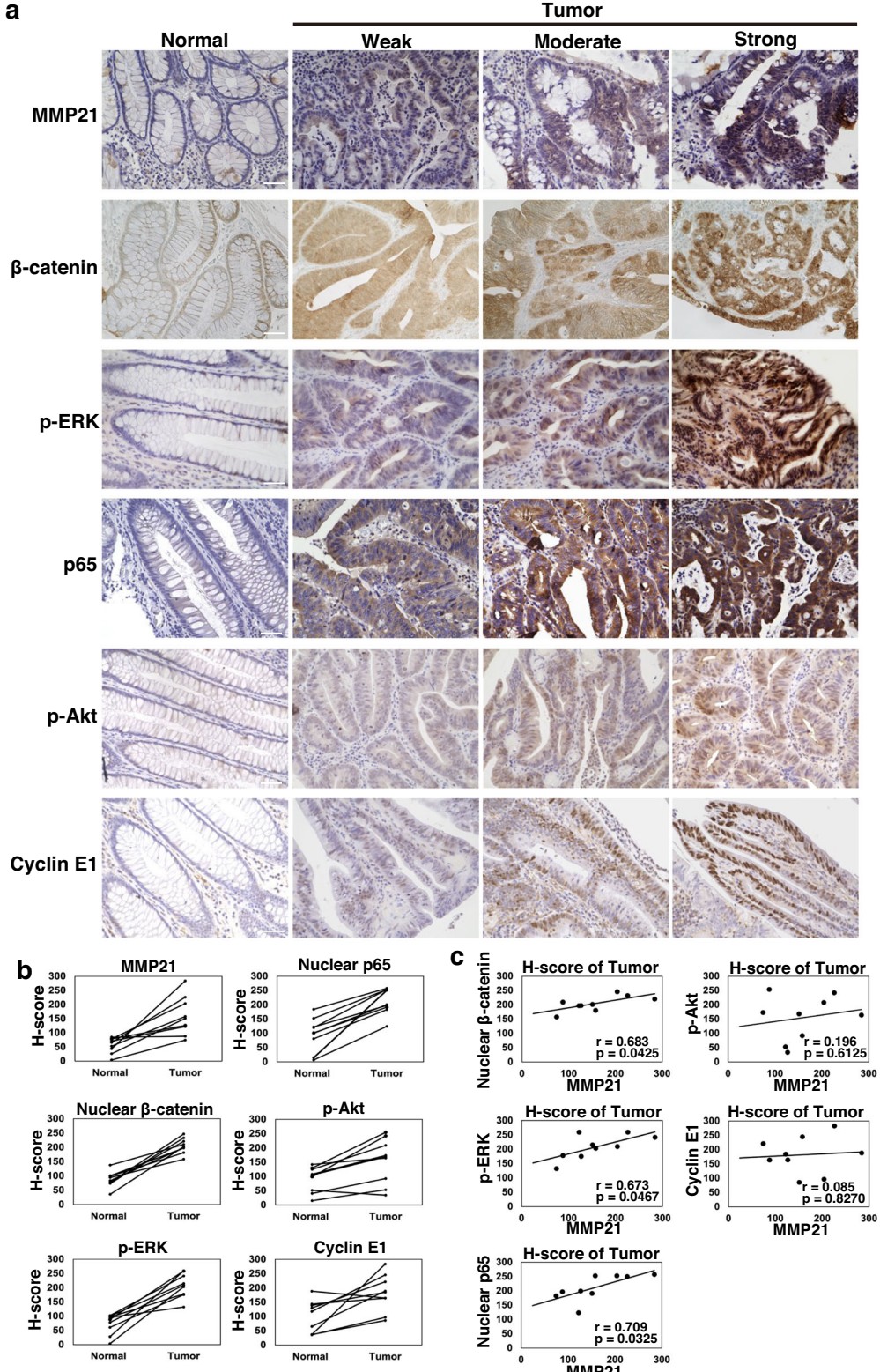

**Fig. 6 | Expression of MMP21 and nuclear β-catenin, p-ERK, or nuclear p65 positively correlate in early colon cancer. a** Representative immunostaining patterns of MMP21, β-catenin, p-ERK, p65, p-Akt, or Cyclin E1. Paraffin-embedded sections of clinical samples were stained with MMP21, β-catenin, p-ERK, p65, p-Akt, or Cyclin E1 antibody and were counterstained with hematoxylin or methyl green. Staining intensities were classified as weak, moderate, or strong. Scale bars, 20 μm. **b** Quantified H-scores of MMP21, nuclear β-catenin, p-ERK, nuclear p65, p-Akt, and Cyclin E1 in normal and tumor tissues. $n = 9$ independent patient-derived samples. **c** Correlation analysis between the H-score of MMP21 and that of nuclear β-catenin, p-ERK, nuclear p65, p-Akt, or Cyclin E1 in tumor tissues. Scatter plots of MMP21 H-score and nuclear β-catenin, p-ERK, nuclear p65, p-Akt, or Cyclin E1 H-score are shown. Correlation coefficient r and *P*-values calculated based on a t-distribution with 7 degrees of freedom are also shown.

*loxP*-eGFP mice, 5′-GCCATCCCTTCACGTTAG-3′, 5′-TTCTGAGAAAG ACAGAAGTTA-3′ and 5′-TTCCACTTTGGCATAAGGC-3′ for the *APC^Min^* mice, and 5′-TAGGAGCAAACCCAATCACTAAAG-3′, 5′-TCCCGCCTAT TCTTTCTGCCCAGC-3′ and 5′-ATCAGACAGAACTATGTGTAACTC-3′ for the *MMP21 KO* mice. The expected sizes of PCR products were 220 bp for *Villin-Cre^ERT2^*, 403 bp for *DNMT1*-CAG-*loxP*-STOP-*loxP*-HRas^V12^-IRES-eGFP, 390 bp for CAG-*loxP*-STOP-*loxP*-eGFP, 619 bp and 331 bp for *APC^Min^*, and 565 bp for *MMP21 WT* allele and 531 bp for *MMP21 KO* allele. For culturing intestinal organoids, isolated crypts from mouse small intestines were entrapped in Matrigel (Corning) and plated in a non-coated 35-mm glass bottom dish[72]. The crypts embedded in Matrigel were covered with Advanced DMEM/F12 supplemented with N2 (Invitrogen), B27 (Invitrogen), 50 ng ml⁻¹ EGF (Peprotech), 100 ng ml⁻¹ Noggin (Peprotech), 1.25 mM N-Acetylcysteine (Sigma) and R-spondin conditioned medium collected from 293T-HA-Rspol-Fc cells kindly provided by Dr. Calvin Kuo (Stanford University). After 96 h culture, organoids were incubated with 100 nM or 1 µM tamoxifen (Sigma) for 24 h to induce transgenes. Subsequently, tamoxifen was washed out, and organoids were cultured for the indicated times. For in vivo experiments, 6–10 weeks old *Villin-RasV12*, *APC^Min^-Villin-RasV12*, or *APC^Min^-Villin-GFP* mice were given a single intraperitoneal injection of 1 mg of tamoxifen in corn oil (Sigma), and were then sacrificed at the indicated days after Cre activation. For inhibition of NF-κB signaling, mice were intraperitoneally injected with 5 mg kg⁻¹ of SN50 daily. To induce intestinal injury, a single dose of indomethacin (12.5 mg kg⁻¹, dissolved in 5% NaHCO₃) was subcutaneously administered to cause epithelial injury in the small intestine. Control mice were injected with the vehicle alone (5% NaHCO₃). To evaluate Wnt activity, small intestinal tissues were collected 8 days after administration. The collected tissues were digested by shaking at 37 °C for 1 h with a mixture of 0.25 mg ml⁻¹ Liberase TM (Roche), 2000 U ml⁻¹ DNase I (Sigma-Aldrich), 10% FBS, and 10 mM HEPES in RPMI 1640 medium (Nacalai Tesque, Inc.). The digested tissues were then passed through a 40 µm cell strainer to obtain single cells. The cell suspensions were subsequently resuspended in 2 mM EDTA, 0.5% BSA in PBS, and CD326 (EpCAM) MicroBeads (mouse) (Miltenyi Biotec) at a concentration of 10 µl per $1 \times 10^7$ cells were added to them, followed by incubation at 4 °C for 15 min. After collecting the cells, EpCAM-positive epithelial cells were isolated using LS columns (Miltenyi Biotec). Total RNA was then extracted and subjected to q-PCR analysis.

## Cell culture

MDCK (a kind gift from W. Birchmeier) and MDCK-pTR GFP-RasV12 cells were used in this study[73]. To establish MDCK mCherry-β-catenin Δ131 or MDCK-pTR GFP-RasV12 mCherry-β-catenin Δ131 cells, cDNA of mCherry-β-catenin Δ131 was cloned into *BamHI* and *EcoRI* sites of a PB-EF1-MCS-IRES-Neo vector. MDCK or MDCK-pTR GFP-RasV12 cells were then transfected with PB-EF1-MCS-IRES-Neo-mCherry-β-catenin Δ131 by nucleofection (nucleofector 2b Kit L, Lonza), followed by selection in medium containing 800 µg ml⁻¹ G418 (Invitrogen). MDCK cells stably expressing mCherry-β-catenin Δ131 in a doxycycline-inducible manner were established by transfecting MDCK cells with pPB-TRE3G mCherry-β-catenin Δ131, followed by selection in medium containing 5 µg ml⁻¹ blasticidin (Invitrogen). To establish MDCK-pTR GFP-RasV12 mCherry-β-catenin Δ131 cells stably expressing *MMP21*-shRNAs, each shRNA sequence were cloned into *AgeI* and *EcoRI* sites of a pLKO.1 vector. The following shRNA sequences were used: *MMP21* shRNA1, 5′-CCGGGCATACTGGAAAGTAGTTAACCTCGAGGTTAACTACTTTCCAG TATGCTTTTTG-3′ and 5′-AATTCAAAAAGCATACTGGAAAGTAGTTA ACCTCGAGGTTAACTACTTTCCAGTATGC-3′; *MMP21* shRNA2, 5′-CCG GGGCAATTTCTATTTTTCAAATCTCGAGATTTGAAAAATAGAAATTGC CTTTTTG-3′ and 5′-AATTCAAAAAGGCAATTTCTATTTTTCAAATCTCG AGATTTGAAAAATAGAAATTGCC-3′. MDCK-pTR GFP-RasV12 mCherry-β-catenin Δ131 cells were then infected with lentivirus carrying pLKO.1-

*MMP21* shRNAs and were cultured in the 1 µg ml⁻¹ puromycin-containing medium and subjected to limiting dilution. For tetracycline-inducible MDCK cell lines, 2 µg ml⁻¹ of tetracycline (Sigma) was used to induce expression of RasV12 mutant except for MDCK-pTRE3G mCherry-β-catenin Δ131 cells for which 2 µg ml⁻¹ of doxycycline (Sigma) was used. For immunofluorescence, cells were plated onto collagen gel-coated coverslips. Type-I collagen (Cellmatrix Type I-A) was obtained from Nitta Gelatin and was neutralized on ice to a final concentration of 2 mg ml⁻¹ according to the manufacturer's instructions. The mixture of type-I collagen and Matrigel at a ratio of 1:4 was used for evaluating basal extrusion of β-cat ΔN/RasV12 cells (Figs. 3c, d, k, l, 5f, g and Supplementary Figs. 7c, d, 9f, g).

## Immunostaining and western blotting

For immunofluorescence, MDCK-pTR GFP-RasV12 mCherry-β-catenin Δ131, MDCK-pTR GFP-RasV12 mCherry-β-catenin Δ131 *MMP21* shRNAs, MDCK-pTRE3G mCherry-β-catenin Δ131 cells were mixed with MDCK or MDCK mCherry-β-catenin Δ131 cells at a ratio of 1:50 and plated onto collagen-coated coverslips[73]. The mixture of cells was incubated for 12-24 h, followed by tetracycline or doxycycline treatment for 24 h except for analyses of basal extrusions, which were examined after 48 h of tetracycline addition. Cells were fixed with 4% paraformaldehyde (PFA) in PBS and permeabilized with 0.5% TritonX-100/PBS[74]. Primary antibodies were used at 1:100 or 1:200, and all secondary antibodies were used at 1:200. Alexa-Fluor-647-conjugated phalloidin was incubated for 1 h at an ambient temperature. For immunofluorescence using intestinal organoids, cells grown in Matrigel were incubated with Cell Recovery Solution (Corning) for 4 min before fixation with 4% PFA. After fixation, cells were permeabilized in 0.5% Triton X-100/PBS for 1 h and blocked in 1% BSA/PBS for 1 h. For immunohistochemical examinations of the small intestine, tissues were isolated, fixed with 10% formalin solution for 24 h, and embedded in paraffin or Tissue-Tek O.C.T. compound (Sakura Finetek Japan Co., Ltd.). For paraffin-embedded samples, the continuous 5 µm-thick sections were sliced. The antigen retrieval was carried out by heating slides for 40 min in 10 mM citrate (pH 6.0) solution. To carry out immunofluorescent staining, the sections were blocked with Block-Ace (DS Pharma Biomedical) and permeabilized with 0.1% Triton X-100 in PBS. For DAB staining, after primary and secondary antibody reactions, the samples were developed with DAB (Nichirei Biosciences Inc.) for 10 min, and counterstained with hematoxylin (Sakura Finetek Japan Co., Ltd.) or methyl green (FUJIFILM Wako). Subsequently, sections were dehydrated with alcohol gradient and treated with xylene to render the slice transparent. To conduct HE staining, paraffin-embedded samples were sliced, deparaffinized, and stained with hematoxylin for 3 min and stained with eosin solution (Sakura Finetek Japan Co., Ltd.) for 30 sec. For frozen samples, 10 µm-thick sections were cut on a cryostat. The sections were blocked with Block-Ace and permeabilized with 0.1% Triton X-100 in PBS. Primary or secondary antibodies were incubated overnight at 4 °C or 4 h at ambient temperature, respectively. All primary antibodies were used at 1:200, and secondary antibodies were at 1:500. Whole-mount immunostaining of mouse small intestinal villi was performed as previously described[75]. Briefly, the small intestine of mice was cut out and put longitudinally on dishes to expose the lumen. After several washes with PBS, tissues were pinned on silicon plates and then fixed with 10% Formalin Solution overnight at 4 °C. The samples were then washed with PBS several times and subsequently dehydrated with 10% sucrose in PBS for 2 h, followed by incubation with 20% sucrose and 10% glycerol in PBS overnight at 4 °C afterward. After blocking with 3% donkey serum (Jackson Immunoresearch Laboratories) in 0.5% Triton-X 100 in PBS for 2 h, samples were incubated with the indicated primary antibodies diluted in the blocking solution overnight at 4 °C, followed by secondary antibodies reaction. After washing with PBS, samples were mounted with Mowiol (Calbiochem). Immunofluorescence images of

intestinal tissues and cultured cells were acquired using the Olympus FV1000 system, Nikon A1R system, or KEYENCE BZ-x800. Images were quantified using the MetaMorph software (Molecular Devices) or the ImageJ/Fiji software. For western blotting, primary antibodies were used at 1:1000[17]. The western blotting data were analyzed using ImageQuant LAS-3000 (GE Healthcare). Uncropped scans of all blots are shown in the Source Data file.

## Microarray analysis

$1.2 \times 10^7$ of 10:1 mix culture of MDCK mCherry-β-catenin Δ131 and MDCK-pTR GFP-RasV12 mCherry-β-catenin Δ131 cells or MDCK and MDCK-pTR GFP-RasV12 cells, and a single culture of MDCK-pTR GFP-RasV12 mCherry-β-catenin Δ131 cells or MDCK-pTR GFP-RasV12 cells were cultured in collagen type I-coated 10-cm plastic dishes. After incubation with tetracycline for 24 h, following Accutase (Nacalai Tesque, Inc.) treatment, GFP-positive cells were collected by an analytical flow cytometer FACS Aria II or Aria III (BD Life Sciences). Total RNA was extracted from the isolated cells using ISOGEN II (Nippon Gene Co., Ltd). The analysis of gene expression profiling was performed using the Canine (V2) Gene Expression Microarray, 4 × 44K (Agilent Technologies).

## Reverse transcription and quantitative real-time PCR

MDCK-pTR GFP-RasV12 mCherry-β-catenin Δ131 cells or a 10:1 mix culture of MDCK mCherry-β-catenin Δ131 and MDCK-pTR GFP-RasV12 mCherry-β-catenin Δ131 cells were cultured at a density of $1.2 \times 10^7$ cells on collagen-coated 10 cm dishes (Corning). After incubation with tetracycline for 24 h, GFP-positive β-cat/RasV12 cells were separated with an analytical flow cytometer. Total RNA was extracted from the isolated cells using ISOGENII and reverse transcribed using a PrimeScript II Reverse Transcriptase (TAKARA) and Oligo(dT)15 Primer (TAKARA), Random Primer 80 nmol (TAKARA), Deoxynucleotide Mix, 10 mM Molecular Biology Reagent (Sigma), RNase Inhibitor, Recombinant (Toyobo Life Science). Luna Universal qPCR Master Mix (New England BioLabs) was used to perform q-PCR using Applied Biosystems 7500 Real-Time PCR System (Applied Biosystems) or QuantStudio 1 Real-Time PCR system (Applied Biosystems). The same procedures were conducted to evaluate Wnt-targeted genes using intestinal organoids or epithelial cells collected from the small intestine of mice. We used β-actin as a reference gene to normalize data. The primer pairs used in the above analyses are listed in Supplementary Tables 1 and 2.

## Patient samples

Early colorectal tumor specimens were provided by the Department of Pathology at the University of Tokyo Hospital. We collected 9 formalin-fixed paraffin-embedded clinical samples from patients who were diagnosed with early colorectal cancer and underwent endoscopic treatments by endoscopic mucosal resection (EMR) or endoscopic submucosal dissection (ESD) (sex, male: $n = 6$, female: $n = 3$; age, 70–82 (median = 76)). Clinical data were collected from electronic medical records. All procedures were in accordance with the ethical standards of the responsible committee on human experimentation (institutional and national) and with the Helsinki Declaration of 1964 and later versions. The institutional Review Board of Tokyo University Hospital approved the use of the abovementioned clinical samples (no. 0542-(7)) for this study, and written informed consent was obtained from all patients.

## Reporter assay

To monitor Wnt activity, MDCK, MDCK mCherry-β-catenin Δ131, MDCK-pTR GFP-RasV12, MDCK-pTR GFP-RasV12 mCherry-β-catenin Δ131 or MDCK-pTRE3G mCherry-β-catenin Δ131 cells were co-transfected with TOP FLASH or FOP FLASH and pRL-TK (Renilla luciferase plasmid). After 24 h, cells were trypsinised, and cultured alone or co-cultured with the

indicated cells for 8 h, followed by tetracycline or doxycycline treatment for 16 h. Firefly luciferase activity was measured using Dual-Luciferase Reporter assay (Promega) and normalized by Renilla luciferase activity using SpectraMax iD5 (Molecular Devices). To evaluate the activity of NF-κB signaling, MDCK-pTR GFP-RasV12 mCherry-β-catenin Δ131 cells were transfected with a pGL4.32 reporter plasmid encoding for the firefly luciferase gene under the control of promoters containing tandem NF-κB elements (Promega) along with a pRL-TK vector. Subsequently, cells were cultured alone or co-cultured with MDCK mCherry-β-catenin Δ131 cells in the absence or presence of RIG012 or CU-CPT 4a and subjected to the same procedure above.

## Purification and enzyme assay of recombinant MMP21 catalytic domain

A 474-bp fragment of the human MMP21 (171 a.a.–328 a.a.) containing catalytic domain was amplified by a PCR method with primers (In-Fusion cloning sites are underlined) fwd 5'-GCCGCGCGGCAGCC ATTCTCCAAGAGGACGCTG-3' and rev 5'-GCTTTGTTAGCAGCCGTT AGGAGCCATACAGCTTTTG-3' using the human MMP21 cDNA in the pcDNA3.1 vector (Addgene). The amplified fragment was ligated in frame into the pET21b expression vector (Novagen) using the NEBuilder HiFi DNA Assembly kit (New England BioLabs), thereby adding a C-terminal His$_6$ tag to the protein. The resulting vector was transformed into *E. coli* BL21 (DE3) cells grown in 2YT medium. MMP21 expression was induced by the addition of 0.5 mM isopropyl β-D-thiogalactopyranoside (IPTG) at OD600 = 0.6, followed by further incubation for 12–16 h at 30 °C. *E. coli* cells were collected by centrifugation and lysed with 50 mM Tris buffer (pH7.5) containing 5 mM CaCl$_2$, 100 mM NaCl, 1 mM ZnSO$_4$, and protease inhibitor cocktail (Roche). Inclusion bodies were solubilized in the same buffer containing 6 M GdnHCl and 5 mM DTT, and loaded on Ni-NTA agarose resin (QIAGEN). After washing with buffer containing 15 mM imidazole, the protein-bound via the C-terminal His$_6$ tag was eluted with a 50 mM Tris buffer (pH 7.5) containing 5 mM CaCl$_2$, 100 mM NaCl, and 500 mM imidazole. Refolding of the recombinant protein was achieved by two-step dilution (1:32) into a 50 mM Tris buffer (pH 7.5) containing 10 mM CaCl$_2$, 100 mM NaCl, 1 mM ZnSO$_4$, 0.1% Brij-35, and 10% glycerol at 4 °C. Refolded MMP21 catalytic domain (1, 5, 10 or 20 μg) was incubated with collagen type I (2.175 μg), collagen type IV (1.5 μg), fibronectin (2.4 μg) or laminin (3.33 μg) in 50 mM Tris (pH 7.5) buffer containing 150 mM NaCl, 5 mM CaCl$_2$ and 0.05% Brij-35 for overnight at 37 °C. The samples were analyzed by SDS–PAGE to evaluate the proteolytic activity of the MMP21 catalytic domain.

## ChIP assay

Chromatin immunoprecipitation (ChIP) was performed using the SimpleChIP Plus Enzymatic Chromatin IP Kit (Magnetic Beads, Cell Signaling Technology, 9005S) following the manufacturer's instructions. Briefly, MDCK-pTR GFP-RasV12 mCherry-β-cateninΔ131 cells were treated with or without 10 ng ml$^{-1}$ TNF-α for 2 h. Subsequently, cells were cross-linked with 1% formaldehyde in PBS for 10 min at room temperature and quenched with 2.5 M glycine for 5 min. Nuclei were prepared, and chromatin was incubated with micrococcal nuclease at 37 °C for 20 min, and subjected to sonication using Sonifier 250A (BRANSON) to produce chromatin smears with an average size of 150–900 bp. The soluble chromatin supernatants (Input samples) were immunoprecipitated with the rabbit anti-p65 antibody (Cell signaling) alongside rabbit IgG control (Cell signaling) overnight at 4 °C. The immunocomplexes were then rotationally incubated with ChIP-Grade Protein G Magnetic Beads for 2 h at 4 °C. ChIP DNA was eluted in ChIP elution buffer (IP sample) and subjected to q-PCR analyses using the following primers, 5'-GGCAACCACTGGGTCTGACT-3' and 5'-TCCGTACGCTGTCAAGTA TTTGAA-3'. Cycle threshold (Ct) values were normalized to the 2% input sample (percentage of input = $2\% \times 2^{(C[T] 2\% \text{ Input sample} - C[T] \text{ IP sample})}$).

## Statistics and reproducibility

For data analyses, unpaired two-tailed Student's *t*-tests (Figs. 1b, g, 3b, d, f, g, i, l, 4h, 5b, d, e, g, l, n and Supplementary Figs. 1a, 3b, 4a, b, d, 6b, 7d, 8, 9b, c, e, g, j) were used to determine *P*-values using Microsoft Excel. For GSEA plots, *P*-values were calculated based on a two-sided permutation test. In Fig. 6c, Pearson's correlation coefficient was calculated, and *P*-values were estimated using a *t*-distribution with 7 degrees of freedom. *P*-values <0.05 were considered to be statistically significant. For animal studies, mice were randomly assigned to experimental groups from each genotype, and investigators were not blinded to allocation for data collection and analysis. Representative figures are shown in Figs. 1a, c, e, f, 2a–e, 3a, c, h, j, k, 4a, b, e, f, g, 5c, f, h, i, j, k, m, 6a and Supplementary Figs. 1b, 2a, b, 3a, 4c, 5a, b, 6a, c, 7a–c, 9d, f, h, i. The experiments were repeated at least three times with similar results, except for immunoblot data: Fig. 3a was repeated three times, Figs. 3j and 4e were repeated twice, Supplementary Fig. 6a was repeated once. All immunoblot data are available in a Source Data file.

## Reporting summary

Further information on research design is available in the Nature Portfolio Reporting Summary linked to this article.

## Data availability

All microarray datasets generated in this study have been deposited into GEO under accession numbers GSE217830 (comprising a mixed culture of MDCK mCherry-β-catenin Δ131 and MDCK-pTR GFP-RasV12 mCherry-β-catenin Δ131 cells, compared to a single culture of MDCK-pTR GFP-RasV12 mCherry-β-catenin Δ131 cells: https://www.ncbi.xyz/geo/query/acc.cgi?acc=GSE217830) and GSE236658 (comprising a mixed culture of MDCK and MDCK-pTR GFP-RasV12 cells, compared to a single culture of MDCK-pTR GFP-RasV12 cells: https://www.ncbi.xyz/geo/query/acc.cgi?acc=GSE236658). All source data are provided in a Source Data file with this paper. Source data are provided with this paper.

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

## Acknowledgements

We thank C. Kuo (Stanford University) for the R-spondin-producing cell line. S. Kon was supported by the AMED Practical Research for Innovative Cancer Control 19217462, Japan Society for the Promotion of Science (JSPS) Grant-in-Aid for Scientific Research on (B) 20H03166, JSPS Grant-in-Aid for Scientific Research on Innovative Areas 21H00441, the Princess Takamatsu Cancer Research Fund, the MSD Life Science Foundation, and The Uehara Memorial Foundation.

## Author contributions

K.N., Y.F., and S. Kon conceived and designed the experiments, and K.N. and S. Kon generated most of the data. H.L., S.Y., S. Kitamoto, E.A., M.K., and M.M. assisted immunohistochemical and pathological analyses of mouse tissue samples. S.T. established cell lines used in this study. H.S., J. Koseki, K.I., R.K., and A.O. conducted the GSEA analysis. K.S., J. Kurauchi, H.T., and H.Y. generated the recombinant MMP21 catalytic domain. Y.S., A.E., S.A., Y.H., T.U. assisted immunohistochemical analyses of clinical samples. The manuscript was written by K.N. and S. Kon with assistance from the other authors.

## Competing interests

A.O. is currently a part-time employee of Astellas Pharma Inc. through a cross-appointment system and was previously employed by Takeda Pharmaceutical Company, Ltd. A.O. has reported paid consulting or advisory roles for Ono Pharmaceutical Company Ltd., Craif Inc., and GEXVal Inc., which are not relevant to this study. The remaining authors declare no competing interests.

## Additional information

[1]Division of Cancer Biology, Research Institute for Biomedical Sciences, Tokyo University of Science, Noda, Chiba 278-0022, Japan. [2]Japan Bioassay Research Center, Japan Organization of Occupational Health and Safety, Kanagawa 257-0015, Japan. [3]Department of Molecular Oncology, Graduate School of Medicine, Kyoto University, Kyoto 606-8501, Japan. [4]Division of Microbiology and Immunology, The WPI Immunology Frontier Research Center (IFReC), Osaka University, Osaka 565-0871, Japan. [5]Division of Translational Genomics, Exploratory Oncology Research and Clinical Trial Center, National Cancer Center, Chiba 277-8577, Japan. [6]Division of Tumor Cell Biology and Bioimaging, Cancer Research Institute, Kanazawa University, Kakuma-Machi, Kanazawa 920-1192, Japan. [7]Division of Systems Biology, Nagoya University Graduate School of Medicine, Nagoya 466-8550, Japan. [8]Division of Cell-Free Sciences, Proteo-Science Center, Ehime University, Matsuyama 790-8577, Japan. [9]Faculty of Pharmaceutical Sciences, Tokyo University of Science, Noda, Chiba 278-8510, Japan. [10]Department of Pathology, Nagoya University Hospital, Nagoya 466-8550, Japan. [11]Department of Gastroenterology, Graduate School of Medicine, University of Tokyo, Tokyo 113-8655, Japan. [12]Department of Pathology, Graduate School of Medicine, University of Tokyo, Tokyo 113-8655, Japan. [13]Department of Molecular-Targeting Prevention, Graduate School of Medical Science, Kyoto Prefectural University of Medicine, Kyoto 602-8566, Japan. ✉e-mail: kon44@rs.tus.ac.jp

