## [Peer Review File · Nature Communications]

Wnt activation disturbs cell competition and causes diffuse invasion of transformed cells through NF- κ B-MMP21 pathwayEditorial Note: Parts of this Peer Review File have been redacted as indicated to maintain the confidentiality of unpublished data.

REVIEWER COMMENTS

Reviewer #1 (Remarks to the Author):

Wnt activation induced disturbance of cell competition causes diffuse invasion of transformed cells through upregulation of NF κ B mediated MMP21

This is an extensive manuscript addressing the combined function of activated Wnt and RTK signaling in epithelial cells. The authors use a heterozygous mutant for APC (APC^{Min}), where Wnt signaling is globally activated to conditionally overexpress RasV12 in differentiated cells of the intestinal villi. In parallel they use similar conditions of RasV12 or GFP overexpression in a wild type background. The key observation is that RasV12 cells surrounded by wild type cells are often extruded apically into the intestinal lumen, but rare, single RasV12 cell in the APC^{Min} background are delaminating basally and presumably progress into more aggressive tumors. Surprisingly this basal delamination is rarely observed if RasV12 activation is expanded to include more than 7 cells in consecutive arrangement. This phenomenon is interpreted as “non-autonomous cell competition”. Interestingly both apical and basal extrusion is also reproduced in organoid cultures when less than 4 cell wide groups of APC^{Min}, RasV12 cells are surrounded by APC^{Min} cells. Similarly, the authors compared MDCK cells expressing a constitutive active form of β catenin (β cat Δ) cultured with and without (β cat Δ) cells coexpressing the RasV12 construct and GFP (β cat Δ , RasV12). When β cat Δ RasV12 cells are cocultured with (β cat Δ) cells at low density, the former ones are predominantly delaminating basally. The authors further investigate the bulk transcriptional differences among the co-cultured cells and show that the delaminating cells upregulate MMP21. Accordingly, MMP1 downregulation in the (β cat Δ , RasV12) reduces their ability to delaminate basally. Similarly inactivation of the MMP1 gene in APC^{Min} mice leads to a reduction of intestinal cells overexpressing RasV12 to delaminate basally. Interestingly MMP21 seems to be upregulated in single APC^{Min}, RasV12 cells that orient both basally and apically! (figure 4 F). In the remaining of the experiments the authors show that MMP21 can be regulated by p65 and that P65 is activated in single APC^{Min} RasV12 cells. Pharmacological inhibition of p65 reduces the basal delamination of single intestinal epithelial cells with increased Wnt and Ras signaling. Overall, this is an interesting and extensive manuscript showing that cell communication between APC^{Min} and APC^{Min} RasV12 cells is an important determinant of extrusion or delamination in intestinal epithelial cancers.

Major comments

A controversy is the persistent characterization of the phenotype as “non cell autonomous”. In my view the data are equally consistent with an autonomous function of NF κ B, MMP21, that is inhibited by the surrounding APC^{Min} RasV12 cells. Dilution of these cells by APC^{Min} cells could lead to delamination. Therefore I propose that the authors exchange the repetitive usage on non-cell autonomous to “cell density dependent”.

The experiment in Figure 3h needs some clarification are both cultures treated with BFA?

In figure 4F MMP21 is upregulated in both apically and basally located GFP cells. Yet the

authors argue that it is specifically required for basal delamination. In Fig4H they argue that cells don't delaminate in the MMP21 mutant background. How can they exclude that the cells have not delaminated and left the epithelium? My expectation would be that more GFP cells would be detected stuck at the basal side of the epithelium. Why are they less in 4H? Ext. Figure 6B addressing the specificity of MMP21 for ECM components is impossible to determine without an internal (non-cleavable) standard included. I could interpret that Col type1 and fibronectin are degraded but the rest are not. Alternatively the authors can provide titration data including lower concentrations of MMP21

Ext data Fig 7. It would be more informative to show all data like in 3b.

Reviewer #2 (Remarks to the Author):

The paper: "Wnt activation-induced disturbance of cell competition causes diffuse invasion of transformed cells through upregulation of NF- κ B-mediated MMP21" by Nakai et al. describes the fate of RasV12 cells in the murine intestine in a WT background or in a Apc+/- background. It is found that while in WT context the Ras mutant cells are frequently extruded in the apical lumen while in a haploinsufficiency Apc setting the preferred route is via basal lamina extrusion.

The experiments are well performed and results clearly described. Having an improved understanding of the dynamics of early tumor formation will be critical to improve preventive strategies and early interventions.

However, there are several aspects of this study that temper my enthusiasm and probably can only be partially resolved.

1. The key drawback of the study is that the studied system is highly artificial and is unclear how this relates to actual cancer formation. As the authors acknowledge themselves in last paragraph of the results section: "Cancer histopathology of APCMin 344 -Villin-RasV12 mice does not mimic any pathological classes of regular human colorectal cancer, thereby, it is hard to obtain the corresponding human clinical samples." I could not have put it in a better way myself but this is a clear problem as it greatly reduces the significance. The relevance of the current mechanism needs to be further substantiated. For example, by studying other more relevant mutations.
2. To me it remains unclear if the relevant comparison in experiment in Fig 3E is indeed B-catenin vs B-catenin/RasV12 as the control with WT co-culture + RasV12 is missing. More broadly and more importantly is the observed effect really unique for Wnt activated environment.
3. In the mechanistic part of the manuscript what is missing is what signal is emanating from the Ras mutant or the Apc mutant cells that regulates NF κ B signaling and MMP21 expression. In other words, how do the mutant cells sense the cellular context which leads to unique MMP21 production in the high Wnt environment.
4. The manuscript focusses on heterozygous Apc mutations as a means to activate Wnt in vivo. Would wounding and tissue repair not be a more relevant setting to study high Wnt environment on the competition between distinct mutant lineages? The problem with the Apc models is that Apc has important roles in the cytoskeleton and response to changes in tissue rigidity which could be impacting on the presented results in vivo.
5. Surprisingly, the highly relevant work of the Ben Simons lab that was published last year

has not been cited (Min Kyu Yum et al.) as well as the papers describing remodeling of the intestinal niche by Wnt activation published in the same Nature issue. More generally, the current findings need to be put in context of the wider literature on competition in the intestine.

Specific comments to Referee #1:

This is an extensive manuscript addressing the combined function of activated Wnt and RTK signaling in epithelial cells. The authors use a heterozygous mutant for APC (APC^{Min}), where Wnt signaling is globally activated to conditionally overexpress RasV12 in differentiated cells of the intestinal villi. In parallel they use similar conditions of RasV12 or GFP overexpression in a wild type background. The key observation is that RasV12 cells surrounded by wild type cells are often extruded apically into the intestinal lumen, but rare, single RasV12 cell in the APC^{Min} background are delaminating basally and presumably progress into more aggressive tumors. Surprisingly this basal delamination is rarely observed if RasV12 activation is expanded to include more than 7 cells in consecutive arrangement. This phenomenon is interpreted as “non-autonomous cell competition”. Interestingly both apical and basal extrusion is also reproduced in organoid cultures when less than 4 cell wide groups of APC^{Min}, RasV12 cells are surrounded by APC^{Min} cells. Similarly, the authors compared MDCK cells expressing a constitutive active form of β catenin (β cat Δ) cultured with and without (β cat Δ) cells coexpressing the RasV12 construct and GFP (β cat Δ , RasV12). When β cat Δ RasV12 cells are cocultured with (β cat Δ) cells at low density, the former ones are predominantly delaminating basally. The authors further investigate the bulk transcriptional differences among the co-cultured cells and show that the delaminating cells upregulate MMP21. Accordingly, MMP1 downregulation in the (β cat Δ , RasV12) reduces their ability to delaminate basally. Similarly inactivation of the MMP1 gene in APC^{Min} mice leads to a reduction of intestinal cells overexpressing RasV12 to delaminate basally. Interestingly MMP21 seems to be upregulated in single APC^{Min}, RasV12 cells that orient both basally and apically! (figure 4 F). In the remaining of the experiments the authors show that MMP21 can be regulated by p65 and that P65 is activated in single APC^{Min} RasV12 cells. Pharmacological Inhibition of p65 reduces the basal delamination of single intestinal epithelial cells with increased Wnt and Ras signaling.

Overall, this is an interesting and extensive manuscript showing that cell communication between APC^{Min} and APC^{Min} RasV12 cells is an important determinant of extrusion or delamination in intestinal epithelial cancers.

We thank the reviewer for acknowledging our work and providing constructive criticism and suggestions. The manuscript has been substantially revised, and we have addressed all the points raised by the reviewer.

1. A controversy is the persistent characterization of the phenotype as “non cell autonomous”. In my view the data are equally consistent with an autonomous function of *Nfkb*, *MMP21*, that is inhibited by the surrounding *APC^{Min} RasV12* cells. Dilution of these cells by *APC^{Min}* cells could lead to delamination. Therefore I propose that the authors exchange the repetitive usage on non-cell autonomous to “cell density dependent”.

We appreciate the comment. We acknowledge that the phrase “non-cell autonomous” might be confusing for some readers. We intended to consistently use this terminology based on several findings in this study;

1. Our observations demonstrate that *APC^{Min}/RasV12* cells (or β -cat Δ N/*RasV12* cells) undergo basally extruding only when surrounded by single gene-mutated cells (*APC^{Min}* cells or β -cat Δ N cells, respectively).

2. We have identified key molecular events (such as *MMP21* expression and NF- κ B signaling) that are crucial for the basal delamination of transformed cells, and these events are potentiated in *APC^{Min}/RasV12* cells (or β -cat Δ N/*RasV12* cells) upon contact with *APC^{Min}* cells (or β -cat Δ N cells).

These findings clearly demonstrate that the observed phenomena are dependent not on cell-intrinsic properties but on the multiplicity of the surrounding *APC^{Min}* cells or β -cat Δ N cells. Therefore, we conclude that Wnt-activated *RasV12*-transformed cells are basally delaminated “non-cell autonomously”. However, we agree with the reviewer that this non-cell autonomous event can also be interpreted as “cell-density dependent”, as the above-mentioned phenotypes disappear when more transformed cells dominate the space. Thus, we have substantially rewritten the text accordingly in response to the reviewer’s proposal. When appropriate, we have replaced “non-cell autonomous” with “cell-density dependent”. These revisions can be found on page 8, line 32, page 10, line 22 and 27, page 11, line 30, page 12, line 14 and 20.

2. The experiment in Figure 3h needs some clarification are both cultures treated with BFA?

We apologize for any lack of clarity regarding certain experiments. When examining *MMP21* expression *in vitro*, BFA was treated in both culture conditions (a mixed or single culture). We have taken care to explicitly describe this point in the manuscript to provide a detailed account of the experimental procedure (on page 9, line 32).

3. In figure 4F MMP21 is upregulated in both apically and basally located GFP cells. Yet the authors argue that it is specifically required for basal delamination. In Fig4H they argue that cells don't delaminate in the MMP21 mutant background. How can they exclude that the cells have not delaminated and left the epithelium? My expectation would be that more GFP cells would be detected stuck at the basal side of the epithelium. Why are they less in 4H?

We appreciate the reviewer for the insightful comment. We have never observed the specific phenotype mentioned by the reviewer, where basally extruded cells get stuck just underneath the epithelial layer, both *in vitro* and *in vivo*. Based on our observations, we believe that MMP21 functions by degrading basement membrane components, enabling transformed cells to breach the basement membrane barrier and become displaced from the epithelia. This assumption is supported by the fact that MMP21-knockout APC^{Min}/RasV12 cells remain within the epithelia (Fig. 4g). Recent studies also suggest that inhibition of MMP activity hinders the basal delamination of *Src*- or *Scribble*-mutated cells induced by cell competition, resulting in their retention within the epithelia, which aligns with our findings (PMID: 35809567, 27997825).

Additionally, as the reviewer noted, MMP21 is upregulated not only in basally extruded cells but also in cells that remain within the epithelia or even apically extrude. This suggests that MMP21 elevation occurs upon exposure to a competitive environment, regardless of the specific cell fate (apical or basal extrusion) that transformed cells undergo. In other words, MMP21 upregulation is not a secondary intracellular event following basal delamination but rather a priming factor that promotes, though not sufficient for, this process. Therefore, we believe that diffusively invading cells are generated stochastically, and not all transformed cells with high MMP21 expression undergo basal delamination. We have now included a more extensive discussion on these points on page 10, lines 27-32 in the revised manuscript. We hope that our message is clearer.

4. *Ext. Figure 6B addressing the specificity of MMP21 for ECM components is impossible to determine without an internal (non-cleavable) standard included. I could interpret that Col typeI and fibronectin are degraded but the rest are not. Alternatively the authors can provide titration data including lower concentrations of MMP21*

We acknowledge the reviewer's criticism and thank the suggested experiments. Following the reviewer's advice, we conducted titration analyses. We incubated varying amounts of the recombinant protein of MMP21 catalytic domain, ranging from 0 μ g to 20 μ g, with substrates. As a result, MMP21 efficiently digests collagen type I and fibronectin, generating both small and large fragments, while collagen type IV is extensively degraded in a dose-dependent manner. In contrast, the degradation of laminin by MMP21 catalytic is less pronounced. These results pinpoint MMP21 as a matrix metalloprotease with unique specificity for certain substrates. Accordingly, we have replaced the data (Supplementary Fig. 7b) and made edits to our manuscript (page 10, lines 6-8, and page 15, lines 8-9).

5. Ext data Fig 7. It would be more informative to show all data like in 3b.

In light of the reviewer's suggestion, we have substituted the original data with figures depicting the values of TOP FLASH and FOP FLASH of β -cat Δ N/RasV12 cells cultured alone or co-cultured with β -cat Δ N cells (Supplementary Fig. 8). The data demonstrate that there are no significant changes in Wnt activity between these conditions.

Specific comments to Referee #2:

The paper: "Wnt activation-induced disturbance of cell competition causes diffuse invasion of transformed cells through upregulation of NF- κ B-mediated MMP21" by Nakai et al. describes the fate of RasV12 cells in the murine intestine in a WT background or in a Apc \pm background. It is found that while in WT context the Ras mutant cells are frequently extruded in the apical lumen while in a haploinsufficiency Apc setting the preferred route is via basal lamina extrusion.

The experiments are well performed and results clearly described. Having an improved understanding of the dynamics of early tumor formation will be critical to improve preventive strategies and early interventions.

However, there are several aspects of this study that temper my enthusiasm and probably can only be partially resolved.

We would like to express our gratitude to the reviewer for dedicating time and effort to reviewing our manuscript. We sincerely appreciate the thoughtful input provided by the reviewer. In response to reviewer's suggestion, we have revised the manuscript.

1. The key drawback of the study is that the studied system is highly artificial and is unclear how this relates to actual cancer formation. As the author acknowledge themselves in last paragraph of the results section: " Cancer histopathology of APCMin 344 -Villin-RasV12 mice does not mimic any pathological classes of regular human colorectal cancer, thereby, it is hard to obtain the corresponding human clinical samples." I could not have put it in a better way myself but this is a clear problem as it greatly reduces the significance. The relevance of the current mechanism needs to be further substantiated. For example, by studying other more relevant mutations.

We appreciate the reviewer's constructive and positive feedback. We agree that the relationship between the cancer pathology of APCMin-Villin-RasV12 mice and actual cancer formation in human remains unclear. In accordance with the reviewer's suggestion, we have evaluated the expression of other colorectal cancer-related genes, specifically p-Akt and Cyclin E1, which serve as indicators of PI3K signal activation and FBXW7 mutation respectively. These genetic alterations are known to frequently occur in relatively advanced stages of colorectal cancers (PMID:31622622, 34217244). As a result, while there was an increase in the H-scores of each protein in tumor samples compared to normal tissues, we did not observe a positive correlation of H-scores between each protein and MMP21. These results suggest that neither activation of PI3K signaling nor FBXW7 mutation is relevant to the induction of MMP21. Instead, they highlight the importance of the NF- κ B-MMP21 pathway in early colorectal cancers caused by Wnt- and MAPK-activation. We have added the new data to Fig. 6 and made corresponding text edits from page 13, line 34 to page 14, line 2.

2. To me it remains unclear if the relevant comparison in experiment in Fig 3E is indeed B-cateN vs B-catN/RasV12 as the control with WT co-culture + RasV12 is missing. More broadly and more importantly is the observed effect really unique for Wnt activated environment.

We appreciate the reviewer's valuable comment. In Fig. 3e, we conducted microarray analysis to identify differentially expressed genes in β -cat Δ N/RasV12 cells during cell competition with β -cat Δ N cells. We thereby compared the gene expression profiles of β -cat Δ N/RasV12 cells co-cultured with β -cat Δ N cells to those of β -cat Δ N/RasV12 cells cultured alone. This analysis led us to identify the NF- κ B-MMP21 pathway as a critical axis for sustaining the competitive benefit of mutated clones. However, as suggested by the reviewer, it remained unclear whether Wnt-activated background was necessary for enhancing this pathway. To address this question, we have performed microarray analysis on RasV12-single transformed cells cultured alone and RasV12 cells co-cultured with parental MDCK cells (RasV12 cells : MDCK cells=1:10). The data from this analysis have been deposited into GEO (accession Number: GSE236658). Consequently, we have found no significant elevation in MMP21 in RasV12 cells co-cultured with MDCK cells. In addition, RIG-I or TLR3-mediated innate immune response leading to NF- κ B signal activation was negligible (Please see Figure for reviewers below). This result indicates that Wnt activation confers the unique phenotype of RasV12-transformed cells in a competitive setting, characterized by NF- κ B-MMP21 activation and basal delamination. We have provided a detailed description of these findings on page 11, lines 20-27.

[Redacted]

3. In the mechanistic part of the manuscript what is missing is what signal is emanating from the Ras mutant or the Apc mutant cells that regulates NfKB signaling and MMP21 expression. In other words, how do the mutant cells sense the cellular context which leads to unique MMP21 production in the high Wnt environment.

We thank for the comment, and agree that this is a very important issue to address. To dissect the underlying mechanism, we have re-analysed the microarray data (Fig. 3e and f) and performed enrichment analysis with the aim of resolving the upstream pathways involved in enhancing NF- κ B signaling. As a result, we have noticed substantial activation of the innate immune systems, particularly the retinoic acid inducible gene-I (RIG-I) and Toll-like Receptor (TLR) signaling pathways (Supplementary Fig. 9a). The expression of RIG-I (also known as DDX58) was profoundly higher in β -cat Δ /RasV12 cells in a competitive environment. Additionally, Toll-Like Receptor 3 (TLR3) showed the highest upregulation among the TLR family members. Both RIG-I and TLR3 are well-known pattern recognition receptors that sense intracellular DNA/RNA products and enhance the innate immune response through NF- κ B signaling (PMID; 25880109, 29677479). Based on these findings, we have investigated the relationship between RIG-I or TLR3 signaling and the NF- κ B-MMP21 pathway. Through q-PCR analysis, we confirmed that RIG-I and TLR3 were significantly upregulated in β -cat Δ /RasV12 cells co-cultured with β -cat Δ cells relative to β -cat Δ /RasV12 cells cultured alone (Supplementary Fig. 9b). Furthermore, we found that inhibition of RIG-I or TLR3 signaling markedly suppressed both NF- κ B signaling activation and MMP21 elevation, suggesting that the activation of these molecules may converge on a common mechanism leading to NF- κ B activation followed by MMP21 expression (Supplementary Fig. 9c, d, e). Additionally, we observed that the cell fates of transformed cells drastically changed upon inhibitor treatment, with a substantial reduction in basal extrusion frequency and promotion of apical extrusion, mimicking the effect of NF- κ B inhibition (Supplementary Fig. 9f, g). Similar effects were observed in organoid cultures (Supplementary Fig. 9h-j). Collectively, these results strongly suggest that the competitive environment entails a RIG-I/TLR3-mediated innate immune response for Wnt-activated RasV12-transformed cells, leading to basal infiltration through the NF- κ B-MMP21 pathway. These results are described from page 12, line 27 to page 13, line 16. We hope that these findings serve

to clarify the upstream events of NF- κ B-MMP21 pathway. The next question would be to determine the ligand for these pattern-recognition receptors, which remains to be characterized mechanistically but undoubtedly deserves future investigation.

4. *The manuscript focusses on heterozygous Apc mutations as a means to activate Wnt in vivo. Would wounding and tissue repair not be a more relevant setting to study high Wnt environment on the competition between distinct mutant lineages? The problem with the Apc models is that Apc has important roles in the cytoskeleton and response to changes in tissue rigidity which could be impacting on the presented results in vivo.*

We highly appreciate the thoughtful comment. We agree with the reviewer that APC mutation could impact the behavior of transformed cells through mechanisms other than Wnt activation. We had thoroughly discussed this issue in the original manuscript (page 14, lines 15-28), and argued that the Wnt-activated condition is causative of the basal delamination of APC^{Min}/RasV12-transformed cells through the non-cell autonomous activation of the NF- κ B-MMP21 pathway. To strengthen our conclusion, as suggested by the reviewer, we have sought to mimic Wnt activation using an alternative method. For this purpose, we have induced severe intestinal injury by subcutaneous injection of indomethacin, a non-steroidal anti-inflammatory drug (PMID: 30718924). We then confirmed that indomethacin profoundly reduced the body weight of mice until 3 days after injection, followed by recovery (Supplementary Fig. 4a). Wnt activity was markedly enhanced 8 days after administration (Supplementary Fig. 4b), and at that time, the basal extrusion of RasV12-transformed cells was significantly promoted (Supplementary Fig. 4c, d), comparable to that observed in the APC^{Min} mice. These results substantiate our conclusion that Wnt activation shifts the balance in cell competition-induced cellular extrusion, favoring the basal delamination of transformed cells. These findings are described from page 6, lines 27-36. Detailed methods are shown from page 18, line 24 to page 19, line 1.

5. *Surprisingly, the highly relevant work of the Ben Simons lab that was published last year has not been cited (Min Kyu Yum et al.) as well as the papers describing remodeling of the intestinal niche by Wnt activation published in the same Nature issue. More*

generally, the current findings need to be put in context of the wider literature on competition in the intestine.

We apologize for overlooking those papers that are obviously germane to this work, as they provide compelling evidence of APC-induced crypt competition. These papers investigated the interplay between transformed and normal crypts, demonstrating that Wnt-activated crypts suppress the expansion of the surrounding normal crypts, acting as a tumor-promoting mechanism. We appreciate the reviewer for bringing these early works to our attention. We have now included a brief discussion of these findings in the Discussion section from page 15, line 35 to page 16, line 3.

REVIEWERS' COMMENTS

Reviewer #1 (Remarks to the Author):

The reviewers have addressed all my concerns

Reviewer #2 (Remarks to the Author):

The manuscript has been significantly improved. Most of my concerns have been addressed.

The relying on Cyclin E / pAKT as a measure for One final question would be to substantiate the generality of findings to PIK3CA mutations/activation or FBXW7 mutations is not the strongest. If at all possible I would suggest to perform mutation analyses on these samples.

Specific comments to Referee #1:

The reviewers have addressed all my concerns.

We thank the reviewer for the assistance with our manuscript.

Specific comments to Referee #2:

The manuscript has been significantly improved. Most of my concerns have been addressed.

We would like to express our gratitude to the reviewer for dedicating time and effort to reviewing our manuscript.

The relying on Cyclin E / pAKT as a measure for One final question would be to substantiate the generality of findings to PIK3CA mutations/activation or FBXW7 mutations is not the strongest. If at all possible I would suggest to perform mutation analyses on these samples.

We sincerely appreciate the thoughtful comment provided by the reviewer. We acknowledge that further evidence in support of the gene mutations on these pathways would be desirable. However, we feel that it is a heavy demand, as the whole picture on genetic alterations occurring at the initial phase of carcinogenesis still remains unclarified, and these issues might reach beyond our scope in this study and should be addressed in future. Moreover, our finding that MMP21 expression is positively correlated with NF- κ B activation, but not with p-Akt activation or Cyclin E1 expression in early colorectal cancers with Wnt- and MAPK-activation provides supportive evidence for implicating the positive role of the NF- κ B-MMP21 pathway in early tumor progression. Therefore, we have adjusted the tone and rephrased the relevant sections by focusing on pathway activities, especially NF- κ B-MMP21 pathway, rather than on genetic alterations in early colorectal cancers. Accordingly, we have made edits from page 13, lines 32 to 35.